# Continual Test-Time Adaptation by Leveraging Source Prototypes and Exponential Moving Average Target Prototypes

## Abstract

Continual Test-Time Adaptation (CTA) is a challenging task that aims to adapt a source pre-trained model to continually changing target domains. In the CTA setting, the model does not know when the target domain changes, thus facing a drastic change in the distribution of streaming inputs during the test-time. The key challenge is to keep adapting the model to the continually changing target domains in an online manner. To keep track of the changing target domain distributions, we propose to maintain an exponential moving average (EMA) target prototype for each class with reliable target samples. We exploit those prototypes to cluster the target features class-wisely. Moreover, we aim to align the target distributions to the source distribution by minimizing the distance between the target feature and its corresponding pre-computed source prototype. We empirically observe that our simple proposed method achieves reasonable performance gain when applied on existing CTA methods. Furthermore, we assess the adaptation time between existing methodologies and our novel approach, demonstrating that our method can gain noteworthy performance without substantial adaptation time overhead.

## 1 Introduction

Despite the huge success of deep neural networks (DNNs) in the field of computer vision and natural language processing, they still suffer from the *distribution shifts* problem which points out the distribution of data given at test time is different from that of the training data. This is because the DNNs heavily rely on the assumption that the data given at test time have the same distribution as the training data. However, it is unlikely in real-world scenarios where the data distribution may change over time due to external factors such as weather change and sensor degradation (Hendrycks & Dietterich, 2019; Koh et al., 2021) at test-time. When the model faces such out-of-distribution (OOD) data during the testing phase, it often fails to generalize and its performance deteriorates. For the sake of clarity, we denote the in-distribution (ID) training data as the source domain data and the OOD test data as the target domain data.

Test-time adaptation (TTA) (Sun et al., 2020a; Wang et al., 2020; Zhang et al., 2022) resolves the distribution shifts problem by adapting the model to the target data given at test-time. Since the target data are unlabeled, the adaptation should be done in an unsupervised and online manner. In an online adaptation scenario, test samples arrive sequentially and the model has to predict and adapt immediately upon the arrival of the test samples. It does not have an access to the full test data but only to the current batch of data, thus prediction of test samples must not be affected by the samples arriving later. Also, TTA must not alter the training procedure of the source domain as done in TTT (test-time training) methods (Sun et al., 2020b; Liu et al., 2021c) since re-training on the source domain is costly. TTA generally assumes that the access to the source data during test-time is infeasible due to privacy/storage concerns and legal constraints, hence the only available during the test time is the access to the target data and the off-the-self source pre-trained model. Recently, another line of research in TTA called continual test-time adaptation (CTA) (Wang et al., 2022; Niu et al., 2022) is introduced. Different from the conventional TTA setting which assumes adapting a model to a single fixed stationary target domain, CTA adapts a model to continually changing target data distributions. The model is not aware of when the input distribution changes since the information about domain change is not given. Therefore, unlike conventional TTA, resetting the

model to the initial source pre-trained weights when the target domain changes is not possible. It is an extremely difficult task resembling the real-world scenarios where the input distribution may change continually and abruptly without prior notice (e.g. entering a tunnel during autonomous driving). Due to its intricate nature, CTA faces two major hurdles. Firstly, it is susceptible to confirmation bias Arazo et al. (2020) issue, wherein the model is overfitted to make incorrect predictions and tends to preferentially predict certain classes more frequently than others. It makes its prediction highly biased and mis-calibrated. The second is the catastrophic forgetting (Kirkpatrick et al., 2017; Parisi et al., 2019; Ebrahimi et al., 2020) of source domain knowledge due to the inaccessibility of the source data at test-time which could impair the performance on source domain as well as the target domains. To alleviate these issues, Wang et al. (2022) proposed augmentation-averaged prediction distillation from the weight-averaged teacher model with stochastic restoration and Niu et al. (2022) proposed sample-efficient optimization strategy with anti-forgetting weight regularization. Please refer to Appendix A for more description about relevant works.

On top of these existing methods, and to further address the two challenges, we present a pair of straightforward yet highly effective techniques: the exponential moving average (EMA) target domain prototypical loss and source distribution alignment via prototype matching. The EMA prototypical loss maintains a prototype for each class by continuously updating the prototypes via EMA using the features of reliable target samples provided during test time. These EMA target prototypes are then utilized to organize the target features into distinct classes. More precisely, we use the pseudo-labels of the reliable target samples to draw their features closer to their corresponding EMA prototypes while simultaneously pushing them away from other irrelevant prototypes through cross-entropy loss. The EMA prototypical loss serves the purpose of effectively capturing the changing target distribution and leveraging it for class-specific target feature clustering. The goal is to prevent an undue bias towards previous target distributions and, instead, adeptly capture and adapt to upcoming target distributions, thereby mitigating the confirmation bias issue.

On the other hand, to tackle the catastrophic forgetting of source domain knowledge as well as to align the target domain distribution to the source, we minimize the distance between the target feature and the pre-computed source prototype. Aligning the distribution between source and target is a very common strategy in domain adaptation (Sun & Saenko, 2016b; Tzeng et al., 2017; Long et al., 2018) which has also been employed in TTA methods (Su et al., 2022). Nonetheless, it relies on the Gaussian distribution assumption of the source and the target domains and utilize complex distance metric such as KL-Divergence. In contrast, our method takes a simpler approach: we directly match each target feature to its corresponding source prototype by minimizing the mean squared error. Similar to the EMA target prototypical loss, we calculate loss only with the reliable target samples. Since CTA requires adaptation in an online manner, adaptation processing time is important as well as the performance. Our investigation illustrates that TTAC (Su et al., 2022) experiences notable fluctuations in adaptation time depending on the domain, whereas our suggested method maintains consistent time for each target domain with insignificant adaptation time.

Our proposed method is both straightforward and efficient, seamlessly applicable to existing approaches without introducing additional parameters or requiring access to the source domain data at test-time (post-deployment). This feature transforms it into a plug-and-play component that can substantially improve the CTA performance without significant adaptation time overhead. Through comprehensive experimentation on ImageNet-C and CIFAR100-C, we establish the compatibility of our method with other CTA methods, showcasing performance improvements in both accuracy and adaptation time. Moreover, we conduct an in-depth analysis of our proposed method, highlighting its capability to mitigate the bias of model to predict certain classes more frequently and enhance ability to make more diverse predictions across a wider range of classes

## 2 PROBLEM DEFINITION

Given a model, $g_{\theta_0}$, pre-trained on a source domain $D^s = \{(x_n^s, y_n^s)\}_{n=1}^{N^s}$, CTA is a task of adapting $g_{\theta_0}$ to the unlabeled target data which its domain continually changes, $D^k = \{x_m^k\}_{m=1}^{N^k}$ ($k$ refers to the target domain index) with an unsupervised objective, $\mathcal{L}_{unsup}$. The target domain data arrive sequentially and their domain changes over time ($k = 1, \ldots, K$). The model only has access to the data of the current time step and has to predict and adapt instantly upon the arrival of the inputs for future steps, $i.e.$, $\theta_t \rightarrow \theta_{t+1}$. As mentioned earlier, the model is not aware of when the target domain changes, so it has to deal with suddenly changing input distribution. $\mathcal{L}_{unsup}$ can take the

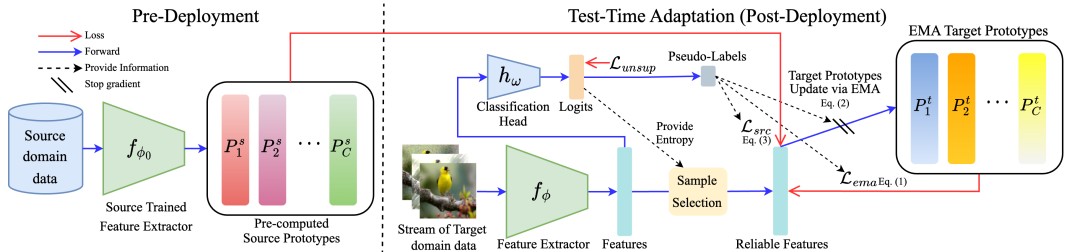

Figure 1: Overall schematic of our proposed test-time adaptation scheme. Left side of vertical dotted line portrays the pre-computation of the source prototypes before deployment. The right side describes the training with the proposed loss terms at test-time. The black dotted arrow indicates providing required information such as entropy and pseudo-label of input.

form of entropy minimization loss which is used to optimize only the affine parameters of batch normalization layer (Wang et al., 2020; Niu et al., 2022) or distillation loss to optimize the whole parameters (Wang et al., 2022; Döbler et al., 2023). The evaluation of the model is determined by test-time predictions in an online manner.

## 3 PROPOSED METHOD

### 3.1 EMA TARGET DOMAIN PROTOTYPICAL LOSS

EMA target prototypical loss comprises two distinct steps, one is categorizing the features of target inputs by classes utilizing the EMA target prototypes and the other is updating the prototypes with features of reliable target samples given at test time in an exponential moving average manner. A classification model, $g_\theta$, consists of a feature extractor $f_\phi$ and a classification head $h_\omega$. Each weight vector $\omega_c \in \mathbb{R}^d$ in $\omega \in \mathbb{R}^{C \times d}$ can be considered as the template for class $c$ where $C$ is the number of classes and $d$ is the dimension of the extracted feature, $f_\phi(x) \in \mathbb{R}^d$. Therefore, we initialize the EMA target prototypes as the weights of $h$, hence $P_c^t = \frac{\omega_c}{\|\omega_c\|_2}$. $P_c^t$ and $\omega_c$ refer to the EMA target prototype and the head weight of class $c$, respectively. We normalize $\omega_c$ to eliminate the difference in magnitudes between $\omega_c$ and the extracted target feature $f_\phi(x^t)$ for updating the target prototypes at the later phase via Eq. (2). There are $C$ EMA target prototypes, which we utilize to categorize the streaming target inputs into classes. This is achieved by minimizing the cross-entropy loss using the pseudo-labels.

However, before proceeding with this process, we first identify reliable target samples as proposed in (Niu et al., 2022), which excludes samples with high entropy, thus low confidence. Given a batch of target data, $\mathbf{x}^t \in \mathbb{R}^{B \times C \times H \times W}$, for each sample $x^t$ in $\mathbf{x}^t$, we calculate its entropy estimated by the model $g_\theta$, *i.e.*, $H(x^t) = -\sum_c^C (\sigma(g_\theta(x^t))_c \cdot \log(\sigma(g_\theta(x^t))_c))$ where $\sigma$ is the softmax operation and $g_\theta(x^t)_c$ refers to the $c$-th element of the produced logit $z^t = g_\theta(x^t) \in \mathbb{R}^C$. Then, we filter out samples with entropy higher than the pre-defined entropy threshold, $E_0$. The remaining samples are the reliable samples with low-entropy denoted as $\tilde{\mathbf{x}}^t$. For each sample $\tilde{x}^t$ in $\tilde{\mathbf{x}}^t$, we obtain its pseudo-label $\tilde{y}^t = \arg\max_c g_\theta(\tilde{x}^t)_c$ and compute the following loss:

$$\mathcal{L}_{ema}(f_\phi, \tilde{x}^t, \tilde{y}^t, P^t) = -\log\left(\frac{\exp(f_\phi(\tilde{x}^t) \cdot \frac{P_{\tilde{y}^t}^t}{\|P_{\tilde{y}^t}^t\|_2})}{\sum_c^C \exp(f_\phi(\tilde{x}^t) \cdot \frac{P_c^t}{\|P_c^t\|_2})}\right). \tag{1}$$

We dot-product $f_\phi(\tilde{x}^t)$ with every EMA target prototype $P_c^t$ and apply softmax operation, then maximize its similarity with the target prototype of the pseudo-label, $P_{\tilde{y}^t}^t$, by minimizing $\mathcal{L}_{ema}$. $\mathcal{L}_{ema}$ assures $f_\phi(\tilde{x}^t)$ to have high similarity with $P_{\tilde{y}^t}^t$ and low similarity with other remaining $P^t$s. $\mathcal{L}_{ema}$ is designed to back-propagate only to the $f_\phi$ and not to the $P^t$s. Upon computing $\mathcal{L}_{ema}$, we proceed to update $P^t$s in an EMA manner using the features of reliable samples and their pseudo-labels as outlined below:

$$P_{\tilde{y}^t}^t = \alpha \cdot P_{\tilde{y}^t}^t + (1 - \alpha) \cdot \frac{f_\phi(\tilde{x}^t)}{\|f_\phi(\tilde{x}^t)\|_2}. \tag{2}$$

---

**Algorithm 1** The pseudo code of our proposed adaptation process for $K$ number of target domains.

---

**Require:** $K$ number of target domains $\{D^k = \{x_m^k\}_{m=1}^{N^k}\}_{k=1}^K$, the source pre-trained model $g_{\theta_0}(\cdot)$,
  Source domain sub-samples $D^s = \{x_n^s\}_{n=1}^{N^s}$, batch size $B$.
1: Pre-compute the source prototype for each class, $P_c^s = \frac{1}{N_c^s} \sum_{i=1}^{N_c^s} f_{\phi_0}(x_i^s)$.
2: Initialize each EMA target prototype, $P_c^t$ as $\frac{\omega_c}{\|\omega_c\|_2}$
3: **for** a domain $k$ in $K$ **do**
4:  **for** a batch $\mathbf{x} = \{x_b^k\}_{b=1}^B$ in $D^k$ **do**
5:   Forward the batch and make predictions, $\mathbf{z} = g_\theta(\mathbf{x})$
6:   Compute $\mathcal{L}_{unsup}$
7:   Compute $\mathcal{L}_{ema}$ and $\mathcal{L}_{src}$ only with the features of reliable target inputs.
8:   Update $P^t$s via Eq. (2)
9:   Optimize model by minimizing Eq. (4).
10:  **end for**
11: **end for**
**Ensure:** The predictions $\{\{\hat{y}_m^k\}_{m=1}^{N^k}\}_{k=1}^K$ for $\{\{x_m^k\}_{m=1}^{N^k}\}_{k=1}^K$

---

Here, $\alpha$ is the blending factor. When the new target features are blended, they are detached to stop gradient signal to $f_\phi$. We also normalize the target feature ($\frac{f_\phi(\tilde{x}^t)}{\|f_\phi(\tilde{x}^t)\|_2}$) as we normalized $\omega_c$ when initializing $P_c^t$. If there exists $N_c$ number of samples with the same pseudo-label in a batch, we use the average of their features ($\frac{1}{N_c} \sum_{i=1}^{N_c} f_\phi(\tilde{x}_i^t)$) instead when updating the target prototype, $P_c^t$. As new batches of target data steam in, $P^t$s are updated with features of new incoming target data in an EMA fashion. The individual magnitudes of each $P^t$ can vary, potentially leading to inaccuracies in the results. To address this issue and ensure consistency in magnitudes, we normalize each $P_c^t$ before performing the dot product with $f_\phi(\tilde{x}^t)$ as described in Eq. (1). Please note that $\mathcal{L}_{ema}$ is computed first, followed by the update of $P^t$ using Eq. (2) with $f_\phi(\tilde{x}^t)$, not the other way around. Also, it is important to mention that $P^t$s are not employed to classify the target input for model evaluation but solely for calculating the loss $\mathcal{L}_{ema}$. The model evaluation is measured by $z = g_\theta(x^t)$, with the head of the model, $h$. It is different from T3A (Iwasawa & Matsuo, 2021) which builds an actual classifier for evaluation with features of target samples given at test-time.

In short, Eq. (1) organizes the target feature into separate classes by enhancing its similarity with the corresponding EMA prototype guided by the pseudo-label while Eq. (2) updates class-specific prototypes gradually and consistently in an EAM manner with the features of target data to reflect the current distribution of test data. The aim of the proposed approach is to quickly adapt to the evolving target distribution by capturing the target distribution well via the EMA target prototypes and utilizing them to cluster the target inputs by classes.

## 3.2 SOURCE DISTRIBUTION ALIGNMENT VIA PROTOTYPE MATCHING

Prior to deploying the model to the target domain for testing, we pre-compute the source prototype for each class in advance using the subset of the source domain data and the feature extractor $f_{\phi_0}$ of the source domain pre-trained model $g_{\theta_0}$. More precisely, we sample a maximum of 100,000 data from the source domain train set. A source prototype for class $c$ is computed as an average of features extracted by $f_{\phi_0}$, hence $P_c^s = \frac{1}{N_c^s} \sum_{i=1}^{N_c^s} f_{\phi_0}(x_i^s)$, where $N_c^s$ is the number of samples with class label $c$ in the subset. Therefore, there exists $C$ prototypes generated with source features before test-time and they are saved in memory to be used later at the adaptation phase. During the test-time adaptation phase, we minimize the mean squared error (MSE) distance between the target feature and the source prototype corresponding to the pseudo-label of the target sample.

$$\mathcal{L}_{src}(f_\phi, \tilde{x}^t, \tilde{y}^t, P^s) = \|P_{\tilde{y}^t}^s - f_\phi(\tilde{x}^t)\|_2^2. \tag{3}$$

Similar to EMA target prototypical loss, we calculate the above source distribution alignment loss only with the reliable samples, $\tilde{\mathbf{x}}^t$. The purpose of $\mathcal{L}_{src}$ is two-fold: firstly, to mitigate the problem of catastrophic forgetting of knowledge from the source domain, and secondly, to align the distribution of target features with that of the source domain in order to reduce the impact of domain shift.

## 3.3 Overall Objective

The overall objective of our proposed test-time adaptation is as follows :

$$\mathcal{L}_{overall} = \mathcal{L}_{unsup} + \lambda_{ema}\mathcal{L}_{ema} + \lambda_{src}\mathcal{L}_{src} \tag{4}$$

$\mathcal{L}_{unsup}$ represents the unsupervised loss employed in the particular method to which our proposed approach is being applied. Our suggested loss components, $\mathcal{L}_{ema}$ and $\mathcal{L}_{src}$, can be integrated into existing methods with respective trade-off terms, $\lambda_{ema}$ and $\lambda_{src}$. Alternatively, they can be employed independently as well, without the inclusion of $\mathcal{L}_{unsup}$. Fig. 1 illustrates our proposed method consists of two strategies each leveraging the prototypes of the source and the target domains. The pseudo code of our test-time adaptation scheme is summarized in Alg. 1.

## 4 Experiments

**Datasets and models.** We evaluate our proposed method on two widely used test-time adaptation benchmarks, CIFAR100-C (Krizhevsky et al., 2009) and ImageNet-C (Deng et al., 2009). Both datasets process (corrupt) the test/validation set of each original dataset with 15 different kinds of corruptions with 5 different levels of severity from four different categories (noise, blur, weather, digital) (Hendrycks & Dietterich, 2019). We conduct experiments with the highest severity of level 5. Other than these 15 corrupted target domains, we also perform test-time adaptation on the original clean test/validation set by setting it as the very last domain to validate how the model has preserved performance on the source domain. We employ ResNeXt29-32×4d pre-trained by Aug-Mix (Hendrycks et al., 2019) and ResNet50 pre-trained by (Hendrycks et al., 2021) as the source pre-trained models for CIFAR100-C and ImageNet-C, respectively. Both models are trained on the original training set of CIFAR-100 and ImageNet.

**Evaluation.** At the very beginning of the test-time adaptation, the model is initialized with the source domain pre-trained weights. As the test-time adaptation initiates, batches of target data stream into the model sequentially for prediction and adaptation. The target domain changes when the model encounters all samples of the current target domain, but the model remains unaware of the domain change. We report the average classification accuracy of 3 runs for each domain.

**Implementation Details.** As our proposed method is compatible with existing approaches, we adhere to the implementation specifics of each method to which our method is applied, including the choice of optimizer and hyper-parameters. To ensure a fair comparison, we conducted all experiments using a consistent batch size of 64 across all methods. The entropy threshold, $E_0$ is set to $0.4 \times \ln C$ following (Niu et al., 2022). $\alpha$, $\lambda_{ema}$ and $\lambda_{src}$ are empirically set to 0.996, 2.0 and 50 when applied on existing method. However, for RMT (Döbler et al., 2023), we use different $\lambda_{ema}$ and $\lambda_{src}$ since it employs source-replay loss which requires source domain data at test-time. More implementation details are in Appendix B. When our proposed method is employed independently without integration into existing methods, $\lambda_{src}$ is set to 20 and we use SGD with a learning rate of 0.00025, momentum of 0.9 and update only the batch normalization layers as done in previous works (Wang et al., 2020; Niu et al., 2022).

### 4.1 Performance Comparison

**Comparison of performance on CTA benchmarks**. To demonstrate the effectiveness of our proposed method, we conducted a comparative analysis with existing methods on two continual test-time adaptation benchmarks. We highlight the versatility of our approach through two distinct strategies: firstly, by integrating it into existing methods, and secondly, by employing the proposed loss terms independently without $\mathcal{L}_{unsup}$ (referred to as **Ours**-Only). Specifically, we applied our proposed terms to three CTA methods, namely EATA, CoTTA, and RMT, which have demonstrated promising performance on the two benchmarks. **Ours** in Tab. 1 and 2 refers to using our proposed terms $\mathcal{L}_{ema}$ and $\mathcal{L}_{src}$ together. As illustrated in the tables, our proposed method shows noteworthy performance when used solely without $\mathcal{L}_{unsup}$ and also significantly enhances performance when incorporated into existing methods. In addition to the aforementioned comparisons, we also assess the performance of our method in comparison to TTAC (Su et al., 2022) and TSD (Wang et al., 2023). They are not originally designed for CTA but have been included as baseline algorithms because their proposed ideas align closely with the philosophy underlying our approach. TTAC places a strong emphasis on domain alignment between the source and target domains by minimizing the Kullback-Leibler (KL) divergence between the distributions of the two domains, under the assumption that each domain follows a Gaussian distribution. While TTAC shares a similar motivation with

Table 1: Classification accuracy (%) for the comparison of continual test-time adaptation performance on ImageNet-C using the highest corruption severity level 5.

| Method | Gauss. | shot | impulse | defocus | glass | motion | zoom | snow | frost | fog | bright. | contrast | elastic. | pixelate | jpeg | original | Mean |
|---|---|---|---|---|---|---|---|---|---|---|---|---|---|---|---|---|---|
| Source | 2.21 | 2.93 | 1.85 | 17.92 | 9.82 | 14.79 | 22.50 | 16.88 | 23.31 | 24.42 | 58.94 | 5.44 | 16.96 | 20.61 | 31.65 | 76.13 | 21.65 |
| T3A | 15.03 | 15.61 | 16.09 | 16.05 | 16.16 | 17.79 | 20.66 | 22.32 | 23.48 | 25.81 | 29.12 | 28.16 | 29.32 | 30.62 | 31.23 | 33.71 | 23.20 |
| TENT | 24.69 | 32.81 | 32.72 | 24.28 | 26.03 | 30.29 | 37.89 | 30.40 | 28.46 | 36.51 | 49.58 | 18.16 | 32.99 | 35.68 | 30.60 | 49.94 | 32.56 |
| TTAC | 23.47 | 32.33 | 32.88 | 24.52 | 29.82 | 40.00 | 47.73 | 42.58 | 40.00 | 50.16 | 61.72 | 26.64 | 47.73 | 51.43 | 45.27 | 66.49 | 41.42 |
| TSD | 15.23 | 15.78 | 15.78 | 15.06 | 15.29 | 26.29 | 38.81 | 34.35 | 33.14 | 47.89 | 65.16 | 16.83 | 44.03 | 48.82 | 39.82 | 75.15 | 34.21 |
| **Ours**-Only | 32.88 | 40.98 | 39.78 | 29.84 | 32.18 | 39.04 | 45.79 | 42.35 | 41.54 | 52.42 | 63.15 | 43.74 | 52.51 | 56.88 | 52.86 | 69.39 | **45.96** |
| EATA | 34.66 | 40.40 | 39.39 | 34.08 | 34.99 | 46.51 | 52.82 | 50.33 | 45.83 | 59.12 | 67.27 | 45.17 | 57.13 | 59.99 | 55.46 | 73.80 | 49.81 |
| EATA + TTAC | 35.64 | 41.44 | 40.57 | 35.59 | 37.14 | 48.67 | 54.56 | 51.69 | 46.73 | 60.34 | 67.98 | 46.58 | 58.04 | 61.22 | 56.18 | 74.40 | 51.05 |
| EATA + **Ours** | 36.17 | 41.77 | 40.83 | 35.98 | 37.24 | 48.89 | 54.28 | 52.15 | 45.79 | 60.23 | 67.94 | 48.01 | 58.26 | 61.26 | 56.37 | 74.20 | **51.32** |
| CoTTA | 16.15 | 18.53 | 19.91 | 18.52 | 19.58 | 31.13 | 43.07 | 36.92 | 36.15 | 51.18 | 65.35 | 23.50 | 47.71 | 52.17 | 44.82 | 73.99 | 37.42 |
| CoTTA + **Ours** | 30.06 | 37.51 | 36.72 | 26.86 | 30.65 | 42.34 | 49.64 | 47.53 | 44.15 | 56.65 | 67.13 | 37.73 | 55.98 | 59.81 | 54.68 | 73.17 | **46.91** |
| RMT | 28.45 | 36.07 | 36.39 | 29.83 | 29.00 | 35.22 | 39.58 | 40.04 | 36.08 | 49.35 | 54.02 | 36.67 | 48.62 | 52.28 | 48.65 | 66.63 | 41.68 |
| RMT + **Ours** | 29.60 | 37.85 | 38.26 | 31.60 | 30.98 | 36.46 | 40.56 | 42.06 | 38.24 | 46.31 | 54.19 | 38.02 | 50.73 | 53.24 | 51.24 | 65.14 | **42.78** |

Table 2: Classification accuracy (%) for the comparison of continual test-time adaptation performance on CIFAR100-C using the highest corruption severity level 5.

| Method | Gauss. | shot | impulse | defocus | glass | motion | zoom | snow | frost | fog | bright. | contrast | elastic. | pixelate | jpeg | original | Mean |
|---|---|---|---|---|---|---|---|---|---|---|---|---|---|---|---|---|---|
| Source | 27.02 | 32.00 | 60.64 | 70.64 | 45.91 | 69.19 | 71.21 | 60.53 | 54.18 | 49.70 | 70.48 | 44.91 | 62.79 | 25.29 | 58.77 | 78.90 | 55.14 |
| T3A | 28.10 | 36.47 | 59.70 | 67.25 | 43.91 | 67.07 | 69.93 | 57.42 | 50.83 | 45.34 | 69.55 | 44.13 | 58.64 | 23.52 | 55.77 | 76.82 | 53.40 |
| TENT | 58.13 | 62.58 | 61.43 | 73.82 | 61.24 | 71.67 | 73.73 | 67.09 | 68.39 | 61.85 | 74.80 | 71.27 | 66.98 | 70.13 | 61.51 | 77.13 | 67.61 |
| TTAC | 58.86 | 63.63 | 61.46 | 72.89 | 59.45 | 70.86 | 72.74 | 65.13 | 66.56 | 59.76 | 73.24 | 68.46 | 63.48 | 67.28 | 60.36 | 75.83 | 66.25 |
| TSD | 56.87 | 58.64 | 56.23 | 71.62 | 57.45 | 69.56 | 71.31 | 64.22 | 64.44 | 57.38 | 72.88 | 68.83 | 63.41 | 66.06 | 58.07 | 75.32 | 64.52 |
| **Ours**-Only | 60.62 | 66.08 | 64.45 | 73.79 | 62.52 | 71.79 | 74.23 | 67.98 | 69.29 | 65.34 | 73.91 | 72.15 | 67.04 | 70.55 | 62.09 | 75.66 | **68.59** |
| EATA | 59.91 | 63.92 | 62.45 | 73.15 | 61.17 | 71.30 | 73.71 | 67.59 | 68.17 | 63.40 | 75.20 | 72.06 | 66.55 | 70.53 | 62.13 | 77.65 | 68.06 |
| EATA + TTAC | 62.28 | 65.54 | 65.59 | 71.90 | 59.06 | 69.63 | 72.13 | 66.00 | 66.47 | 63.38 | 72.97 | 69.55 | 63.85 | 69.06 | 60.82 | 75.21 | 67.09 |
| EATA + **Ours** | 61.29 | 65.66 | 65.32 | 74.31 | 62.79 | 72.41 | 74.77 | 69.16 | 69.95 | 65.99 | 76.22 | 73.76 | 67.75 | 71.78 | 63.42 | 77.99 | **69.53** |
| CoTTA | 59.53 | 62.34 | 60.73 | 72.02 | 62.37 | 70.48 | 72.09 | 65.86 | 66.73 | 59.08 | 72.97 | 69.69 | 65.16 | 69.20 | 63.89 | 74.28 | 66.65 |
| CoTTA + **Ours** | 60.22 | 63.06 | 62.35 | 73.23 | 62.37 | 71.40 | 73.85 | 68.84 | 68.51 | 61.79 | 75.03 | 71.93 | 66.07 | 70.68 | 63.43 | 76.62 | **68.09** |
| RMT | 62.70 | 65.69 | 64.74 | 74.54 | 67.16 | 73.98 | 76.05 | 72.87 | 73.40 | 69.66 | 77.42 | 76.11 | 74.24 | 76.23 | 71.79 | 78.25 | 72.18 |
| RMT + **Ours** | 63.21 | 67.33 | 66.86 | 74.81 | 68.47 | 74.30 | 76.11 | 73.56 | 74.07 | 70.87 | 76.94 | 76.42 | 74.79 | 76.47 | 72.93 | 77.58 | **72.79** |

our $\mathcal{L}_{src}$, our approach is designed to be much simpler and more efficient. In the case of TSD, it introduces a concept akin to our $\mathcal{L}_{ema}$, but there is a fundamental difference in that TSD utilizes a memory bank, whereas our method maintains the target prototypes via EMA. As demonstrated in the table, although these methods share some common ideas with our approach, our proposed method consistently outperforms them in the two benchmarks. We have also evaluated performance of TTAC applied to EATA, EATA+TTAC. Its performance on ImageNet-C is comparable to EATA+**Ours**, but it falls slightly short. Moreover, TTAC exhibits significant fluctuation in adaptation time depending on the domain which will be further studied in the next section.

**Adaptation processing time comparison**. Adaptation processing time stands as another important factor to take into account during test-time adaptation. This is particularly important in a CTA scenario, where the model has to predict and adapt immediately in an online manner. Therefore, we present the average time it takes to adapt a single batch for each target domain and compare between methods. The experiment is conducted on a single NVIDIA RTX 3090 GPU with a fixed batch size of 64 for fair comparison. Fig. 2 illustrates the comparison of the average adaptation time of a single batch across the target domains. What stands out is the results of TTAC. Its average adaptation time of a single batch exhibits significant

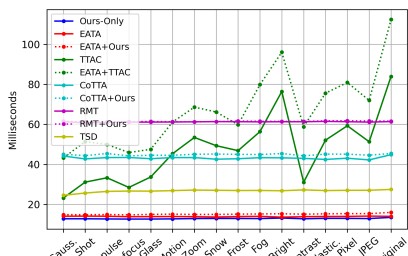

Figure 2: Comparison of average batch adaptation time for each target domain on ImageNet-C.

variability across the target domains. This is attributed to TTAC's calculation of the covariance matrix only using samples with high confidence. It implies that more computational effort is needed for a particular domain which the model predicts with high confidence. On the other hand, **Ours**-Only shows not only consistent adaptation time across the target domains but also the least amount of time required. Even when applied on existing methods such as EATA, CoTTA, and RMT, it incurs only a marginal adaptation time overhead. From the experimental results of Tab. 1 and Fig. 2, we demonstrate that our proposed method is able to boost the test-time adaptation performance only with a negligible amount of adaptation time overhead.

Table 3: The results of random order of target domains on ImageNet-C

| Method | Tent | TTAC | TSD | **Ours**-Only | EATA | EATA+TTAC | EATA+**Ours** | CoTTA | CoTTA+**Ours** | RMT | RMT+**Ours** |
|---|---|---|---|---|---|---|---|---|---|---|---|
| Acc. (%) | 14.50±1.43 | 41.22±0.72 | 34.21±0.01 | 45.98±0.24 | 49.56±0.28 | 50.68±0.22 | 50.91±0.23 | 37.73±0.09 | 46.78±0.17 | 44.72±0.58 | 45.11±0.61 |

Table 4: Ablation study on ImageNet-C

| Time | | | | | $t$ ⟶ | | | | | | | | | | | | | | | | →  | |
|---|---|---|---|---|---|---|---|---|---|---|---|---|---|---|---|---|---|---|---|---|---|---|
| Method | | | | | | | | | | | | | | | | | | | | | | Mean |
| EATA | $\mathcal{L}_{ema}$ | $\mathcal{L}_{src}$ | Normal. | Filter. | Gauss. | shot | impulse | defocus | glass | motion | zoom | snow | frost | fog | bright. | contrast | elastic. | pixel. | jpeg | original | |
| ✓ | – | – | – | – | 34.66 | 40.40 | 39.39 | 34.08 | 34.99 | 46.51 | 52.82 | 50.33 | 45.83 | 59.12 | 67.27 | 45.17 | 57.13 | 59.99 | 55.46 | 73.80 | 49.81 |
| ✓ | ✓ | – | ✓ | ✓ | 35.55 | 41.41 | 40.50 | 35.01 | 36.69 | 47.77 | 53.77 | 51.14 | 46.70 | 59.66 | 67.41 | 45.70 | 57.58 | 60.52 | 55.68 | 73.82 | 50.56 |
| ✓ | – | ✓ | – | ✓ | 35.40 | 41.10 | 40.03 | 35.05 | 36.30 | 48.10 | 53.97 | 51.56 | 46.83 | 59.85 | 68.00 | 47.49 | 57.80 | 60.94 | 56.07 | 74.27 | 50.80 |
| ✓ | ✓ | ✓ | – | – | 36.32 | 41.47 | 40.10 | 34.84 | 35.76 | 47.82 | 53.48 | 51.43 | 46.62 | 59.71 | 67.60 | 47.39 | 57.70 | 60.76 | 55.87 | 74.06 | 50.68 |
| ✓ | ✓ | ✓ | – | ✓ | 36.47 | 41.79 | 40.72 | 35.39 | 36.25 | 48.07 | 53.73 | 51.68 | 46.92 | 59.84 | 67.74 | 47.79 | 57.86 | 60.85 | 56.02 | 74.05 | 50.95 |
| ✓ | ✓ | ✓ | ✓ | – | 36.11 | 41.72 | 40.65 | 35.56 | 36.77 | 48.62 | 54.04 | 51.95 | 47.08 | 60.14 | 67.85 | 47.83 | 58.02 | 61.09 | 56.11 | 74.15 | 51.11 |
| ✓ | ✓ | ✓ | ✓ | ✓ | 36.17 | 41.77 | 40.83 | 35.98 | 37.24 | 48.89 | 54.28 | 52.15 | 47.46 | 60.23 | 67.94 | 48.01 | 58.26 | 61.26 | 56.37 | 74.20 | 51.32 |

**Continual test-time adaption on a random order of target domains**. Since CTA involves adapting to target inputs as they arrive sequentially, the order in which the target domains are presented can significantly impact the model's performance. The original domain sequence consists of consecutive domains within the same categories (noise, blur, weather, digital), making it easier for the model to gradually adapt. In contrast to the original domain sequence, we introduce randomness by shuffling the order of the target domains, including the 15 corrupted target domains within ImageNet-C, with the original source domain placed at the end. This randomization allows us to evaluate the robustness of each method to variations in the order of target domains. We compute the average accuracy across all 16 test domains based on three separate runs, each with a distinct domain order. As shown in Table 3, the results reveal that certain methods exhibit improved performance, while others experience a decrease in performance compared to the original domain sequence. Notably, **Ours**-Only and methods enhanced with our approach demonstrate increased resilience to variations in the order of domains, consistently achieving superior performance when compared to the baseline methods.

## 4.2 ANALYSIS

In the following analysis, all experiments are conducted on ImageNet-C with ResNet50.

**Ablation study**. Tab. 4 presents the results of our ablation study, aimed at confirming the effectiveness of our proposed loss terms and the specific implementation choices we have made. We assess the validity of each component of our proposed method by gradually incorporating them into the baseline algorithm, EATA. The term 'Normal.' in the table refers to normalizing $\omega$ and $f_\phi(\tilde{x}^t)$ when initializing and updating $P^t$, while 'Filter.' indicates the filtering of unreliable samples with high entropy. The second and third rows showcase the validation of our proposed loss terms, as performance improves when each loss term is added. Subsequently, the fourth to sixth rows demonstrate the significance of normalization and reliable sample selection. When both techniques are not used, there is only a modest performance improvement compared to EATA (row 4). The importance of normalization becomes evident as its removal leads to a significant drop in performance (row 5). While filtering also contributes to performance gains, its removal results in a minor performance drop (row 6). This highlights that our proposed method can robustly operate even with unreliable samples possessing high entropy. Overall, the ablation study confirms the effectiveness of our proposed loss terms and specific implementations to the performance improvements.

**Batch size**. While it is a well-established fact that larger batch sizes often result in improved model performance, the TTA setting can not guarantee large batch size as it operates online and requires immediate prediction and adaptation. Therefore, we conduct a performance comparison between EATA and EATA+**Ours** across six different batch sizes (128, 64, 32, 16, 8, 4) to evaluate the robustness of our proposed method to batch size variations. As presented in Fig. 3 (a), it is evident that EATA+**Ours** consistently outperforms EATA from batch size 128 to 16. However, at a batch size of 8, EATA exhibits superior accuracy, and at a batch size of 4, both methods yield poor performance due to the extremely limited number of inputs.

**Blending factor** $\alpha$. The blending factor $\alpha$ governs the extent to which the target prototypes, $P^t$, are updated by the incoming target features. A smaller $\alpha$ promotes quicker update to new features, while a larger $\alpha$ results in a more gradual update of $P^t$, preserving their similarity to their initial states. In Fig. 3 (b), we conduct an analysis of how the performance varies in EATA+**Ours** with different values of $\alpha$ (0.9, 0.96, 0.99, 0.996, 0.999). It is evident that for all five values, EATA+**Ours** outperforms the baseline algorithm EATA (49.81%). The results clearly indicate higher accuracy

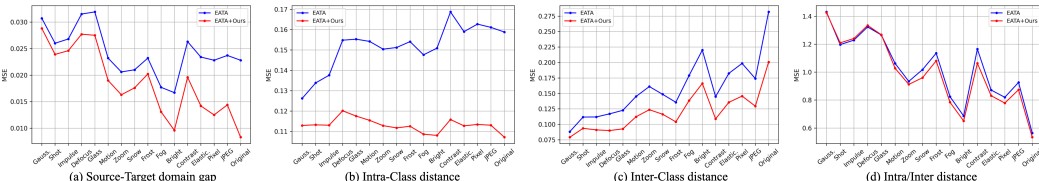

(a) Batch size ablation      (b) $\alpha$ ablation      (c) $\lambda_{ema}$ ablation      (d) $\lambda_{src}$ ablation

Figure 3: Analysis of batch size, $\alpha$, $\lambda_{ema}$ and $\lambda_{src}$ on ImageNet-C. (a) presents a comparison between EATA and EATA+**Ours** with varying batch sizes, while (b), (c), and (d) show performance analysis using different $\alpha$, $\lambda_{ema}$ and $\lambda_{src}$ employed in our method. Accuracy (%) is the average accuracy over the 16 domains (15 corrupted and original).

(a) Source-Target domain gap    (b) Intra-Class distance    (c) Inter-Class distance    (d) Intra/Inter distance

Figure 4: Feature space distance analysis. (a) plots the domain gap between the source and the target. (b) and (c) show the intra-class and the inter-class distance, respectively while (d) presents the ratio (intra/inter) of the two distance.

with larger values of $\alpha$ and the lowest accuracy when $\alpha$ is set to 0.9. This observation suggests that excessive updating of $P^t$ with smaller values of $\alpha$ negatively impacts performance.

**Trade-off terms** $\lambda_{ema}$ **and** $\lambda_{src}$. Fig. 3 (c) and (d) provide an analysis of the trade-off parameters $\lambda_{ema}$ and $\lambda_{src}$ associated with our proposed loss terms $\mathcal{L}_{ema}$ and $\mathcal{L}_{src}$ within the EATA+**Ours** model. When we vary the values of $\lambda_{ema}$, $\lambda_{src}$ is held constant at 50. Conversely, when analyzing $\lambda_{src}$, $\lambda_{ema}$ is set at 2. The model achieves its highest accuracy when $\lambda_{ema}$ is set to 2, with a decline in performance as $\lambda_{ema}$ increases. On the other hand, accuracy shows a gradual increase with rising $\lambda_{src}$ values, peaking at 50. Beyond this value, accuracy does not exhibit significant changes. Although there are differences in accuracy for various values of $\lambda_{ema}$ and $\lambda_{src}$, the gap between the highest and lowest accuracy is relatively small. This suggests that our proposed loss terms are not highly sensitive to the choice of trade-off values.

**Source-Target domain gap**. We analyze the distribution alignment between the source and the target by measuring the mean squared error (MSE) between the source prototypes and the target prototypes computed with the ground-truth (GT) labels. Unlike $P^t$s which are generated with the pseudo-labels, we calculate $P^{t^*}$ with the GT labels. During test-time adaptation, we store the features produced by $f_\phi$ and compute $P^{t^*}$ for each class as follows, $P_c^{t^*} = \frac{1}{N_c^t} \sum_{i=1}^{N_c^t} f_\phi(x_i^t)$ where $N_c^t$ is the number of samples with GT label $c$. For each test domain, we compute the average MSE between $P^s$ and $P^{t^*}$ over the classes, $\frac{1}{C} \sum_{c=1}^{C} \|P_c^s - P_c^{t^*}\|_2^2$. Fig. 4 (a) illustrates the domain gap of EATA and EATA+**Ours**. The notably lower distance observed in EATA+**Ours** compared to EATA across all test domains indicates that our proposed terms contribute significantly to narrowing the distribution gap between the source and the target domains.

**Intra- and inter-class distance of target features**. We also measure the intra-class and inter-class distance to analyze how our proposed method affects class-wise feature distributions. Intra-class distance is the average of the distances between the feature for every input to their corresponding $P^{t^*}$s, $d_c^{intra} = \frac{1}{N_c^t} \sum_{i=1}^{N_c^t} \|P_c^{t^*} - f_\phi(x_i^t)\|_2^2$, which can be used to check how well the features are clustered. Inter-class distance is the average of distances between $P^{t^*}$s of different classes, which is to see how well the clusters are separated, $d_c^{inter} = \frac{1}{C-1} \sum_{i=1}^{C} \mathbb{1}_{\{i \neq c\}} \|P_c^{t^*} - P_i^{t^*}\|_2^2$. We measure both distances for each class and report the average over classes for each target domain. In Fig. 4 (b) and (c), we present a comparison between EATA and EATA+**Ours** for both intra-class and inter-class distances. The intra-class distance for EATA+**Ours** remains consistently lower, whereas for EATA, it gradually increases, leading to a widened gap between the two methods as the adaptation progresses. It validates that our proposed terms contribute in minimizing intra-class variance. On the other hand, concerning inter-class distance, EATA exhibits larger distances than EATA+**Ours**, suggesting that

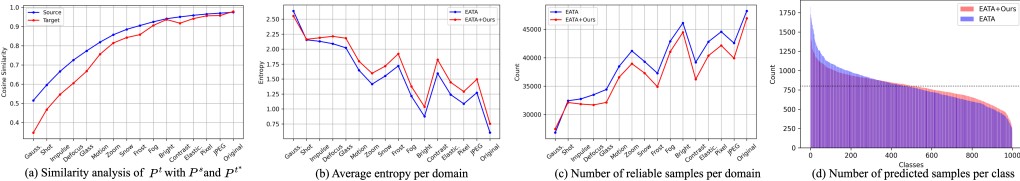

Figure 5: (a) shows the similarity analysis of $P^t$ with $P^s$ and $P^{t^*}$. (b), (c) and (d) illustrate the average entropy per domain, the number of reliable samples and predicted samples per class compared between EATA and EATA+**Ours**, respectively.

the prototypes are more widely dispersed. Nonetheless, it is noteworthy that the gap between the two methods remains relatively constant throughout the target domains when compared to the intra-class distances. It may be tempting to conclude that EATA achieves a more class-discriminative feature distribution due to its higher inter-class distance. However, when we examine the ratio between the two distances ($d_c^{intra}/d_c^{inter}$) in Fig. 4 (d), EATA+**Ours** consistently yields lower values, especially for later target domains. A lower ratio implies a relatively larger inter-class distance compared to the intra-class distance, indicating higher separability.

**Similarity analysis of $P^t$ with $P^s$ and $P^{t^*}$.** $P^t$s play a crucial role in computing $\mathcal{L}_{ema}$. Their significance lies in their enhanced ability to accurately represent the centroids of the class clusters. To assess their representations of the centroids of the class clusters, we analyze their cosine similarity with the prototypes of the source and the target domains ($P^s$s and $P^{t^*}$s) which are constructed with the ground-truth labels, hence the actual centroids of the clusters. As shown in Fig. 5 (a), it is observed that as the test-time adaptation proceeds, $P^t$s gradually show high similarity with both $P^s$s and $P^{t^*}$s. The high similarity suggests that the EMA target prototypes, $P^t$s, accurately represent the actual centroids of the class clusters. Further discussion about it continues in the appendix H.

**Comparison of the number of samples predicted per class.** We find an intriguing observation which the entropy of EATA+**Ours** is higher than EATA in Fig. 5 (b) despite its superior accuracy over EATA. This may appear counterintuitive, as entropy minimization loss is often employed to enhance performance. Also, in Fig. 5 (c) which shows the remaining reliable samples after excluding the unreliable samples, it is observed that EATA+**Ours** retains fewer samples than EATA after filtering, primarily due to its higher entropy. To investigate the reason behind the high entropy in EATA+**Ours**, we plot the number of predicted samples for each class and compare them between the two methods in Fig. 5 (d). These results are obtained using all the samples from the 16 test domains. The classes are sorted in descending order of the number of predicted samples for clarity, with the horizontal dotted line indicating the actual number of samples assigned to each class. Interestingly, EATA predicts certain classes much more frequently than others, resulting in a highly skewed distribution of predicted samples. This implies there is a strong bias in the model that forces it to predict specific classes more often. In contrast, EATA+**Ours** distributes predictions more evenly across the classes compared to EATA. This explains why EATA+**Ours** exhibits higher entropy, as it tries to predict more diversely across the classes. This result validates that our proposed terms alleviate the confirmation bias issue by leading the model to make a more balanced and diverse class predictions which results in improved accuracy. Result of each domain is in appendix I.

## 5 CONCLUSION

This paper proposes simple and efficient method of leveraging the prototypes of the source and the target domains for continual test-time adaptation. Its compatibility with existing CTA methods makes it a simple plug-and-play component that can be easily applied. The source prototypes are employed to minimize the distribution shifts between the source and the target domains while the target prototypes are used to organize the target features into class clusters. The target prototypes are updated in an EMA manner using the target samples given at test-time. Our findings reveal that our proposed terms significantly improve the performance of the model with minimal adaptation time overhead. Furthermore, we observe that our proposed terms alleviate the bias in the model by encouraging it to make a more balanced predictions across classes rather than favoring specific classes. We anticipate that our research can serve as a stepping stone for further investigation and the advancement of more robust continual test-time adaptation methods.

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

# A   RELATED WORKS

## A.1   UNSUPERVISED DOMAIN ADAPTATION

When there is a domain shift, where the distribution of test data differs from that of the training data, model performance tends to degrade (Ganin & Lempitsky, 2015). Many unsupervised domain adaptation (UDA) methods have been extensively researched, adapting models to various domain shifts without additional annotation costs by using labeled source domain data and unlabeled target domain data during training. Following a seminal work Ganin & Lempitsky (2015), several approaches (Zhu et al., 2020; Tang & Jia, 2020; Xu et al., 2020; Chen et al., 2019; Shen et al., 2018) align the feature distribution of both the source and the target domain through adversarial learning or transforming source domain images into the target style to effectively transfer the task knowledge to the target domain (Zhu et al., 2020; Luo et al., 2020). As another research avenue, several methods (Kim et al., 2019; Liu et al., 2021a; Zou et al., 2018) have been proposed where models trained in the source domain are utilized to generate pseudo-labels for the target domain, facilitating self-training to adapt the model to the target domain. Recently, there has been research in source-free domain adaptation (SFDA) (Liang et al., 2020), aiming to address the inefficient aspects of data transferring and privacy leakage issues by adapting models to the target domain using only the unlabeled target domain data, without relying on the source domain data. It is a challenging task since it cannot access the labeled source data while adapting the model to the target domain.

## A.2   TEST-TIME ADAPTATION

Recently, test-time adaptation (TTA) has garnered substantial attention, adapting models to specific test domains during inference-time after being deployed to the target data. TTA shares similarities with SFDA in the aspect of adapting off-the-shelf models pre-trained on source domain to the target domain without accessing source domain data. However, TTA differs from SFDA that it is an online learning approach relying solely on the incoming target samples given at inference-time without repetitively accessing a large amount of unlabeled target domain data, which makes TTA more challenging in that overall information such as knowing the target domain distribution (Sun & Saenko, 2016a) or clustering the target features (Liang et al., 2020) are not available. Many studies (Wang et al., 2020; Niu et al., 2022; Lim et al., 2023) efficiently adapt models to the test domain by updating only the batch normalization layer, following the research (Schneider et al., 2020) that only replacing the statistics for batch normalization without learning can effectively address domain shifts. These methods (Wang et al., 2020; Niu et al., 2022) adapt the model to the target domain via entropy minimization loss so that the predictions inferred by the current model become more confident. Alternatively, there are approaches Su et al. (2022); Jung et al. (2022) that update the entire backbone so that the distribution of the target domain feature has similar statistics to that of the source on the premise that the statistics of the source domain features are known. Some other methods Iwasawa & Matsuo (2021); Jang et al. (2023) entirely freeze the backbone and solely modify the classifier by leveraging prototypes derived from target domain features based on pseudo-labels. Additionally, some methods (Sun et al., 2020a; Bartler et al., 2022) modify the model architecture during source domain training to incorporate self-supervised losses, which are then used as self-supervised losses for the target data during test time.

## A.3   CONTINUAL TEST-TIME ADAPTATION

In practice, the distribution of the test domain can exhibit continuous changes or have correlations among continuously incoming samples, whereas TTA relies on a strong assumption that test-time data follows i.i.d, meaning that the distribution of the test-time data does not change and stays stationary. CoTTA (Wang et al., 2022) first suggests the problem of continual test-time adaptation (CTA) and proposes corresponding problem setting. It identifies the problem of error accumulation in existing TTA methods when the distribution of test-time data changes and addresses it by introducing a teacher-student framework and ensuring that various augmented test samples have consistent predictions along with stochastic restoration of the weights to the source-trained weights. Following this, Brahma & Rai (2023) and Döbler et al. (2023) also utilize a teacher-student structure, employing regularization based on the importance of weights and using symmetric cross-entropy loss, respectively to robustly update the model and prevent catastrophic forgetting. Additionally, Niu et al. (2022), which considers the confidence and diversity of samples for model updates, has proven to

be effective in the context of CTA. Building upon existing TTA methods, Song et al. (2023); Hong et al. (2022) have proposed techniques to diminish memory consumption, thereby promoting efficient adaptation in CTA.

## B IMPLEMENTATION DETAILS

Here, we describe the implementation details of each method in our experiments. We use the code implemented in MECTA Hong et al. (2022)[1] for TENT (Wang et al., 2020), EATA Niu et al. (2022), and CoTTA (Wang et al., 2022). For other methods, we referenced official implementation of each method. We use PyTorchh Paszke et al. (2019) framework and a single NVIDIA RTX 3090 GPU for conducting experiments.

**Tent**. (Wang et al., 2020) We use the SGD optimizer with a learning rate of 0.0001 and a momentum of 0.9 for both ImageNet-C and CIFAR100-C datasets.

**T3A**. (Iwasawa & Matsuo, 2021) We referenced the official code of T3A [2] for its implementation. Since it is an optimization free method, there is no need for an optimizer as well as a learning rate. We use 100 for the hyper-parameter $M$ which indicates the $M$-th largest entropy of the support set.

**TSD**. (Wang et al., 2023) We referenced the official code of TSD [3] for its implementation. We use the ADAM (Kingma & Ba, 2014) optimizer with a learning rate of 0.00005 for both ImageNet-C and CIFAR100-C datasets as mentioned in its paper. We use 3 for the number of nearest neighbors $K$, 100 for the entropy filter hyper-parameter $M$ and 0.1 for the trade-off parameter $\lambda$ following its implementation details described it its paper.

**TTAC**. (Su et al., 2022) We referenced the official code of TTAC [4] for its implementation. We used the implementation version that does not use the queue since saving target data in queue at test-time costs memory and computation overhead which are not suitable for continual test-time adaptation. We use the SGD optimizer with a learning rate of 0.0002/0.00001 and momentum of 0.9 for ImageNet-C and CIFAR100-C datasets, respectively. However, when we apply TTAC on EATA, we follow the implementation details of EATA and use a learning rate of 0.00025 and update only the batch normalization layers. We use 0.9, 0.9, 1280, 64 for $\tau_{PP}$, $\xi$, $N_{clip}$, $N_{clip\_k}$ and 0.05/0.5 for the trade-off parameter of global feature alignment, $\lambda$, in ImageNet-C and CIFAR100-C datasets, respectively, following its official implementation.

**EATA**. (Niu et al., 2022) We use the SGD optimizer with a learning rate of 0.00025 and a momentum of 0.9 for both ImageNet-C and CIFAR100-C datasets. The entropy threshold $E_0$ is set as $0.4 \times \ln C$ as mentioned earlier in the main paper and the threshold for redundant sample identification, $\epsilon$, is set to 0.05. The number of samples for calculating Fisher information is set to 2000 and the trade-off parameter for anti-forgetting loss, $\beta$, is set to 2000 as well for both datasets. The moving average factor to track the average model prediction of a mini-batch for redundant sample identification is set to 0.1 as mentioned in its implementation details.

**CoTTA**. (Wang et al., 2022) We use the SGD optimizer with a learning rate of 0.0001 and a momentum of 0.9 for the ImageNet-C dataset, whereas we employ the ADAM optimizer with a learning rate of 0.001 for CIFAR100-C. The confidence threshold for deciding whether to augment the provided inputs, denoted as $p_{th}$, is configured at 0.1/0.72, while the restore probability for generating masks for stochastic restoration, represented as $p$, is established at 0.001/0.01 for the ImageNet-C and CIFAR100-C datasets, respectively. The exponential moving average momentum for the update of the teacher model is set to 0.999 in both datasets. Originally, CoTTA uses the output of the teacher model for the evaluation, but when we apply our proposed method on CoTTA we use the output of the student for the evaluation. Also, we use the same learning rate of 0.0001 regardless of the datasets when applying our method on CoTTA.

**RMT**. (Döbler et al., 2023) [5] We use the SGD optimizer with a learning rate of 0.01 and a momentum of 0.9 for the ImageNet-C dataset, whereas we employ the ADAM optimizer with a learning rate

---

[1] https://github.com/SonyResearch/MECTA
[2] https://github.com/matsuolab/T3A
[3] https://github.com/SakurajimaMaiii/TSD
[4] https://github.com/Gorilla-Lab-SCUT/TTAC
[5] https://github.com/mariodoebler/test-time-adaptation

Table 5: Classification accuracy (%) for the comparison of continual test-time adaptation performance on ImageNet-C using the corruption severity level 3.

| Time | $t$ | | | | | | | | | | | | | | | | |
|---|---|---|---|---|---|---|---|---|---|---|---|---|---|---|---|---|---|
| Method | Gauss. | shot | impulse | defocus | glass | motion | zoom | snow | frost | fog | bright. | contrast | elastic. | pixelate | jpeg | original | Mean |
| Source | 27.60 | 25.04 | 25.16 | 37.94 | 16.86 | 37.73 | 35.23 | 35.19 | 32.11 | 46.65 | 69.62 | 46.01 | 55.61 | 46.20 | 59.29 | 76.13 | 42.02 |
| T3A | 44.09 | 42.59 | 43.94 | 36.38 | 32.79 | 47.76 | 48.82 | 45.46 | 40.35 | 58.08 | 67.29 | 60.09 | 62.30 | 60.01 | 59.00 | 70.18 | 51.20 |
| TENT | 52.3 | 55.46 | 54.33 | 45.49 | 46.14 | 54.54 | 53.90 | 48.43 | 45.08 | 58.64 | 66.58 | 61.92 | 65.11 | 63.46 | 63.10 | 70.28 | 56.55 |
| TTAC | 52.77 | 56.55 | 54.08 | 41.24 | 43.85 | 54.00 | 51.60 | 47.97 | 43.28 | 56.66 | 63.33 | 50.69 | 58.61 | 56.13 | 56.37 | 64.96 | 53.26 |
| TSD | 44.51 | 42.22 | 43.5 | 36.14 | 31.49 | 49.34 | 50.05 | 46.62 | 40.65 | 61.26 | 71.33 | 63.26 | 65.64 | 63.49 | 62.05 | 75.15 | 52.92 |
| **Ours**-Only | 56.07 | 59.42 | 58.29 | 49.94 | 49.95 | 57.70 | 57.49 | 54.17 | 49.42 | 62.41 | 69.50 | 67.03 | 67.90 | 66.79 | 65.49 | 72.30 | **60.24** |
| EATA | 57.2 | 58.86 | 58.06 | 52.50 | 51.45 | 60.84 | 60.29 | 58.14 | 51.64 | 66.52 | 71.65 | 68.20 | 69.65 | 68.08 | 66.81 | 74.53 | 62.15 |
| EATA + TTAC | 57.78 | 59.86 | 58.97 | 53.57 | 52.38 | 61.67 | 61.10 | 58.87 | 52.24 | 67.02 | 72.20 | 68.55 | 69.97 | 68.45 | 67.01 | 74.80 | 62.78 |
| EATA + **Ours** | 58.23 | 59.97 | 59.16 | 54.10 | 52.73 | 61.87 | 61.09 | 59.52 | 52.96 | 67.11 | 72.14 | 68.86 | 70.05 | 68.68 | 67.26 | 74.93 | **63.04** |
| CoTTA | 45.76 | 45.94 | 48.19 | 40.87 | 37.90 | 54.18 | 54.80 | 49.82 | 44.72 | 62.98 | 71.26 | 65.41 | 67.60 | 65.36 | 64.75 | 74.81 | 55.90 |
| CoTTA + **Ours** | 56.49 | 59.28 | 58.63 | 51.18 | 51.19 | 61.19 | 60.20 | 58.74 | 51.54 | 65.76 | 71.68 | 68.66 | 69.74 | 68.62 | 66.86 | 74.13 | **62.12** |
| RMT | 51.44 | 54.38 | 53.83 | 48.65 | 46.64 | 52.47 | 51.71 | 52.63 | 47.97 | 58.27 | 62.51 | 59.77 | 63.45 | 63.16 | 62.92 | 67.47 | 56.08 |
| RMT + **Ours** | 51.33 | 54.35 | 54.17 | 48.77 | 47.63 | 53.31 | 52.97 | 53.74 | 49.32 | 59.11 | 62.81 | 60.48 | 62.74 | 62.78 | 62.75 | 66.35 | **56.41** |

Table 6: Classification accuracy (%) for the comparison of continual test-time adaptation performance on CIFAR100-C using the corruption severity level 3.

| Time | $t$ | | | | | | | | | | | | | | | | |
|---|---|---|---|---|---|---|---|---|---|---|---|---|---|---|---|---|---|
| Method | Gauss. | shot | impulse | defocus | glass | motion | zoom | snow | frost | fog | bright. | contrast | elastic. | pixelate | jpeg | original | Mean |
| Source | 36.52 | 47.18 | 73.13 | 76.76 | 60.61 | 72.29 | 75.43 | 69.59 | 62.14 | 72.27 | 76.63 | 71.91 | 73.69 | 69.10 | 64.12 | 78.89 | 67.52 |
| T3A | 38.26 | 51.24 | 70.53 | 74.49 | 57.82 | 69.54 | 73.20 | 66.64 | 59.35 | 69.39 | 75.21 | 69.38 | 70.81 | 67.23 | 60.69 | 76.85 | 65.66 |
| TENT | 62.72 | 67.77 | 72.15 | 77.06 | 68.72 | 73.77 | 76.08 | 71.64 | 70.80 | 73.98 | 77.45 | 76.10 | 74.36 | 75.24 | 65.60 | 77.71 | 72.57 |
| TTAC | 63.24 | 68.79 | 71.38 | 76.04 | 66.02 | 71.86 | 73.88 | 68.69 | 67.89 | 70.76 | 74.10 | 71.59 | 70.27 | 70.70 | 61.91 | 74.38 | 70.10 |
| TSD | 61.64 | 64.83 | 68.43 | 74.72 | 65.49 | 71.37 | 73.46 | 68.22 | 67.17 | 70.54 | 74.95 | 72.87 | 71.65 | 72.11 | 62.22 | 75.32 | 69.69 |
| **Ours**-Only | 65 | 70.51 | 73.50 | 77.64 | 69.56 | 74.09 | 76.53 | 72.26 | 71.46 | 74.46 | 76.93 | 76.10 | 74.33 | 75.08 | 66.01 | 76.95 | **73.15** |
| EATA | 63.8 | 68.72 | 71.77 | 76.54 | 68.06 | 73.21 | 75.74 | 71.31 | 70.70 | 73.80 | 77.44 | 75.49 | 74.17 | 74.87 | 65.72 | 77.68 | 72.44 |
| EATA + TTAC | 62.74 | 65.01 | 64.84 | 67.67 | 61.47 | 64.95 | 67.37 | 64.86 | 64.58 | 65.61 | 67.92 | 65.81 | 66.60 | 67.35 | 62.09 | 68.28 | 65.45 |
| EATA + **Ours** | 65.59 | 70.29 | 73.54 | 77.69 | 69.92 | 74.21 | 76.75 | 72.54 | 72.46 | 75.05 | 78.15 | 77.22 | 75.26 | 76.10 | 67.19 | 78.21 | **73.76** |
| CoTTA | 63.32 | 67.13 | 69.73 | 75.64 | 69.14 | 73.31 | 75.06 | 71.43 | 70.37 | 73.19 | 76.06 | 74.94 | 74.08 | 74.99 | 68.93 | 76.39 | 72.11 |
| CoTTA + **Ours** | 64.25 | 68.07 | 71.87 | 76.38 | 69.09 | 73.45 | 75.76 | 72.51 | 71.31 | 74.24 | 77.45 | 75.95 | 74.57 | 75.31 | 67.56 | 77.40 | **72.82** |
| RMT | 66.39 | 70.57 | 73.36 | 77.80 | 73.32 | 76.20 | 77.74 | 75.98 | 76.02 | 77.33 | 78.61 | 78.38 | 78.39 | 78.50 | 75.15 | 78.87 | 75.79 |
| RMT + **Ours** | 66.74 | 71.44 | 74.11 | 77.44 | 74.02 | 76.37 | 77.67 | 76.35 | 76.47 | 77.42 | 78.21 | 78.10 | 77.97 | 78.08 | 75.84 | 78.18 | **75.90** |

of 0.0001 for CIFAR100-C. The number of samples for warm up is set to 50,000 and the trade-off parameters for contrastive loss and the source replay loss are set as 1. The temperature for contrastive loss and the exponential moving average momentum for teacher model update are set to 0.1 and 0.999, respectively. Note that RMT is not a source-free method since is employs source-replay loss during test-time adaptation which requires source domain data even at the test-time. Other than the source replay loss, it also employs contrastive loss which makes the overall loss term of RMT intricate. Therefore, when we apply our proposed terms on RMT, we use different values of $\lambda_{ema}$ and $\lambda_{src}$. For ImageNet-C, we use $\lambda_{ema} = 0.5$ and $\lambda_{src} = 0.01$ while we use $\lambda_{ema} = 1.0$ and $\lambda_{src} = 0.01$ in CIFAR100-C.

We adhere to the hyper-parameters as detailed in the paper or the official implementation of each method. Nevertheless, for some methods, we fine-tuned the learning rate to align with our continual test-time adaptation setting, maintaining a fixed batch size of 64.

## C  RESULTS ON DIFFERENT SEVERITY LEVEL

Table 5 and  6 show the performance comparison between methods with the corruption severity level 3 on the two benchmarks, ImageNet-C and CIFRA100-C under the continual test -time adaptation setting. Even with a lower severity level of corruption, Our proposed terms consistently contribute to the improvement of performance when applied to baseline methods. It is also worth highlighting that **Ours**-only demonstrates significant performance independently as well, without relying on existing methods.

## D  CONSISTENCY LOSS WITH STRONG AUGMENTATION

Employing consistency loss between original input and its augmented version is a widely used technique in semi/self-supervised learning to improve the generalization capacity of the model (Chen

Table 7: Ablation study of consistency loss on ImageNet-C using the corruption severity level 5.

| Time | t → | | | | | | | | | | | | | | | | |
|---|---|---|---|---|---|---|---|---|---|---|---|---|---|---|---|---|---|
| Method | Gauss. | shot | impulse | defocus | glass | motion | zoom | snow | frost | fog | bright. | contrast | elastic. | pixelate | jpeg | original | Mean |
| EATA | 34.66 | 40.40 | 39.39 | 34.08 | 34.99 | 46.51 | 52.82 | 50.33 | 45.83 | 59.12 | 67.27 | 45.17 | 57.13 | 59.99 | 55.46 | 73.80 | 49.81 |
| EATA + **Ours** | 36.17 | 41.77 | 40.83 | 35.98 | 37.24 | 48.89 | 54.28 | 52.15 | 47.46 | 60.23 | 67.94 | 48.01 | 58.26 | 61.26 | 56.37 | 74.20 | 51.32 |
| EATA + **Ours** + $\mathcal{L}_{cons}$ | 36.66 | 42.33 | 41.41 | 36.25 | 37.57 | 48.91 | 54.04 | 52.58 | 47.65 | 60.34 | 67.94 | 48.39 | 58.22 | 61.36 | 56.56 | 74.27 | **51.53** |
| EATA + **Ours** + $\mathcal{L}_{cons}$(CoTTA-Aug) | 35.15 | 40.30 | 39.50 | 33.92 | 35.83 | 47.38 | 53.06 | 51.20 | 46.62 | 59.54 | 67.26 | 47.12 | 57.48 | 60.49 | 55.77 | 73.72 | 50.27 |
| CoTTA | 16.15 | 18.53 | 19.91 | 18.52 | 19.58 | 31.13 | 43.07 | 36.92 | 36.15 | 51.18 | 65.35 | 23.50 | 47.71 | 52.17 | 44.82 | 73.99 | 37.42 |
| CoTTA + **Ours** | 30.06 | 37.51 | 36.72 | 26.86 | 30.65 | 42.34 | 49.64 | 47.53 | 44.15 | 56.65 | 67.13 | 37.73 | 55.98 | 59.81 | 54.68 | 73.17 | 46.91 |
| CoTTA + **Ours** + $\mathcal{L}_{cons}$ | 31.38 | 39.62 | 38.97 | 28.78 | 32.16 | 43.25 | 50.39 | 48.93 | 44.34 | 57.10 | 67.07 | 39.35 | 55.69 | 59.74 | 54.75 | 72.49 | **47.75** |
| CoTTA + **Ours** + $\mathcal{L}_{cons}$(CoTTA-Aug) | 27.57 | 34.75 | 35.07 | 27.60 | 30.50 | 42.37 | 49.56 | 46.66 | 43.31 | 55.85 | 66.73 | 39.35 | 54.70 | 58.77 | 53.22 | 73.19 | 46.20 |

et al., 2020; He et al., 2020; Grill et al., 2020; Liu et al., 2021b; Sohn et al., 2020). Since TTA is also a kind of unsupervised learning, it adopts such strategy as well. CoTTA (Wang et al., 2022) is the first TTA work to propose the use of EMA teacher network and employing the consistency loss between the outputs of the teacher and the outputs of the student with various augmentations on the inputs to the teacher network. However, we find that consistency loss can achieve better performance with stronger augmentation strategy and even without the use of the teacher network.

We do not employ the teacher network and give two versions of input (original and strong augmented version) to the network. Instead of using the augmentations used in CoTTA, we adopts augmentations proposed in Liu et al. (2021b) which employs randomly adding color jittering, grayscale, Gaussian blur, and cutout patches.

$$\mathcal{L}_{cons}(g_\theta, x^t, \mathcal{A}) = -\sum_{c}^{C}(\sigma(g_\theta(x^t)) \cdot \log(\sigma(g_\theta(\mathcal{A}(x^t)))))^c \tag{5}$$

The consistency loss is defined as the cross-entropy loss between the outputs of the two inputs (original and its augmented version) predicted by the same network $g_\theta$ where $\mathcal{A}$ and $\sigma$ refer to the augmentation and the softmax operation. $\mathcal{L}_{cons}$ can be additionally incorporated with a balancing trade-off parameter, $\lambda_{cons}$ which makes the overall objective as follows:

$$\mathcal{L}_{overall} = \mathcal{L}_{unsup} + \lambda_{ema}\mathcal{L}_{ema} + \lambda_{src}\mathcal{L}_{src} + \lambda_{cons}\mathcal{L}_{cons}. \tag{6}$$

We apply the consistency loss to both EATA+**Ours** and CoTTA+**Ours** to demonstrate its effectiveness. Table 7 presents the respective results, clearly indicating that $\mathcal{L}_{cons}$ contributes to performance improvement. Particularly, its impact is more pronounced when applied to CoTTA. However, when we use the augmentation strategies proposed in CoTTA for $\mathcal{A}$, denoted as $\mathcal{L}_{cons}$(CoTTA-Aug) in the table, the performance rather deteriorates. This result emphasizes the importance of using a proper augmentation strategy for the consistency loss. Our experiment suggests that using strong augmentation such as random cutout patches is indeed effective.

Table 8: Ablation study of $\lambda_{ema}$ on **Ours**-Only using the ImageNet-C

| $\lambda_{ema}$ | 1 | 2 | 3 | 4 | 5 |
|---|---|---|---|---|---|
| Acc. (%) | 45.20 | **45.96** | 44.69 | 34.29 | 26.55 |

Table 9: Ablation study of $\lambda_{src}$ on **Ours**-Only using the ImageNet-C

| $\lambda_{src}$ | 10 | 20 | 30 | 40 | 50 | 60 | 70 |
|---|---|---|---|---|---|---|---|
| Acc. (%) | 45.58 | **45.96** | 45.86 | 45.65 | 45.29 | 43.44 | 41.01 |

# E ABLATION STUDY ON TRADE-OFF TERMS $\lambda_{ema}$ AND $\lambda_{src}$ OF **OURS**-ONLY

As mentioned in the implementation details described in Section 4, when our proposed loss terms are used independently without integration into existing methods, we use $\lambda_{ema}$=2 and $\lambda_{src}$=20. Table 8 and 9 show the ablation study of $\lambda_{ema}$ and $\lambda_{src}$ with different values when our proposed terms are solely used without $\mathcal{L}_{unsup}$. When examining the effect of $\lambda_{ema}$, $\lambda_{src}$ is set at 20, whereas when investigating the impact of $\lambda_{src}$, $\lambda_{ema}$ is configured to 2. The accuracy in the tables are an average accuracy over the 16 test domains. Table 8 illustrates that the performance reaches its peak at $\lambda_{ema} = 2$, and it experiences a sharp decline when values exceed 3. Similarly, Table 9 reveals that similar performance is maintained from 10 to 50, achieving over 45% accuracy, but it sharply declines when values surpass 50.

Table 10: Performance comparison between soft label and hard label for $\mathcal{L}_{ema}$ on ImageNet-C

| Time | $t \longrightarrow$ | | | | | | | | | | | | | | | | |
|---|---|---|---|---|---|---|---|---|---|---|---|---|---|---|---|---|---|
| Method | *Gauss.* | *shot* | *impulse* | *defocus* | *glass* | *motion* | *zoom* | *snow* | *frost* | *fog* | *bright.* | *contrast* | *elastic.* | *pixelate* | *jpeg* | *original* | Mean |
| EATA | 34.66 | 40.40 | 39.39 | 34.08 | 34.99 | 46.51 | 52.82 | 50.33 | 45.83 | 59.12 | 67.27 | 45.17 | 57.13 | 59.99 | 55.46 | 73.80 | 49.81 |
| EATA + **Ours** Hard Label | 36.17 | 41.77 | 40.83 | 35.98 | 37.24 | 48.89 | 54.28 | 52.15 | 47.46 | 60.23 | 67.94 | 48.01 | 58.26 | 61.26 | 56.37 | 74.20 | 51.32 |
| EATA + **Ours** Soft Label | 35.89 | 41.60 | 40.80 | 35.72 | 37.30 | 48.82 | 54.33 | 52.07 | 47.42 | 60.28 | 68.04 | 48.05 | 58.35 | 61.29 | 56.34 | 74.31 | 51.29 |

Table 11: Performance comparison between student output and teacher output of CoTTA + **Ours** on ImageNet-C

| Time | $t \longrightarrow$ | | | | | | | | | | | | | | | | |
|---|---|---|---|---|---|---|---|---|---|---|---|---|---|---|---|---|---|
| Method | *Gauss.* | *shot* | *impulse* | *defocus* | *glass* | *motion* | *zoom* | *snow* | *frost* | *fog* | *bright.* | *contrast* | *elastic.* | *pixelate* | *jpeg* | *original* | Mean |
| CoTTA | 16.15 | 18.53 | 19.91 | 18.52 | 19.58 | 31.13 | 43.07 | 36.92 | 36.15 | 51.18 | 65.35 | 23.50 | 47.71 | 52.17 | 44.82 | 73.99 | 37.42 |
| CoTTA + **Ours** Teacher Output | 21.75 | 33.04 | 36.38 | 25.07 | 30.68 | 39.15 | 47.03 | 41.41 | 41.80 | 52.41 | 65.50 | 35.47 | 51.56 | 56.23 | 51.52 | 72.38 | 43.84 |
| CoTTA + **Ours** Student Output | 30.06 | 37.51 | 36.72 | 26.86 | 30.65 | 42.34 | 49.64 | 47.53 | 44.15 | 56.65 | 67.13 | 37.73 | 55.98 | 59.81 | 54.68 | 73.17 | 46.91 |

## F  COMPARISON OF HARD LABEL AND SOFT LABEL FOR $\mathcal{L}_{ema}$

We use the pseudo-label $\tilde{y}^t$ when calculating loss $\mathcal{L}_{ema}$. The pseudo-label can take the form of a one-hot vector, serving as a hard label, or it can be used as the raw logit output of the model, acting as a soft label. When using the soft-label, we minimize the cross-entropy loss between the output of the EMA target prototypes and the soft pseudo-label. The output of the EMA target prototypes refers to a logit, $z_{ema}^t \in \mathbb{R}^C$, produced by dot-producting $f_\phi(x^t)$ with every $P_c^t$ for each class. In the main paper, we present results using the hard label representation. However, to delve deeper into the mechanism of $\mathcal{L}_{ema}$, we perform a performance comparison using both versions of the pseudo-label, as summarized in Table 10. As demonstrated in the table, there is no significant distinction between the two versions of the pseudo-label, although the hard-label version exhibits slightly better performance.

## G  COMPARISON OF STUDENT OUTPUT AND TEACHER OUTPUT OF CoTTA+OURS

As specified in the implementation details, CoTTA originally uses the output of the teacher network for evaluation, but we employ the output of the student network when applying our proposed loss terms on CoTTA. Table 11 presents a performance comparison between CoTTA+**Ours** using the output of the teacher and the output and the student. As demonstrated in the table, using the teacher network's output yields inferior performance compared to the student network's output, yet it still significantly outperforms CoTTA without our proposed terms. We hypothesize that the reason for the student output's superior accuracy is that our proposed loss terms directly impact the student network, whereas the teacher network undergoes slow updates through exponential moving average.

## H  SIMILARITY ANALYSIS OF $P^t$ WITH $P^s$ AND $P^{t^*}$.

Fig. 6 shows the results of our similarity analysis of $P^t$ with $P^s$ and $P^{t^*}$. After the model sees all the samples of a target domain, we measure the cosine similarity between the $P^t$s and the $P^s$s and the $P^t$s and $P^{t^*}$s for the target domain. We report the cosine similarity averaged over the classes, $\frac{1}{C}\sum_{c=1}^{C} cos(P_c^t, P_c^s \, or \, P_c^{t^*})$ where $cos$ denotes cosine similarity. The blue plot shows the similarity with the source prototypes, $P^s$, while the red plot shows the similarity with the target prototypes $P^{t^*}$. Note that $P^{t^*}$s are computed using the ground truth labels, so they represent the actual centroids of the class clusters of the target domains. As shown in the figure, as the adaptation proceeds, the similarity with both the source and the target prototypes increase. It implies that as $P^t$s are slowly updated in an EMA manner with the features of the reliable samples, they better represent the true centroids of the class clusters. We also observe that the similarity with the source prototypes smoothly increases as the adaptation goes on. We conjecture this is due to our proposed source prototype alignment loss $\mathcal{L}_{src}$ which regulates the feature extractor $f_\phi$ to align the target feature distribution to that of the source. Also, the tendency of increasing similarity with the target prototypes, $P^{t^*}$ indicates that even though $P^t$s are updated using the pseudo-label information, since only reliable samples are employed, they succeed in maximizing similarity with the ground-truth prototypes, $P^{t^*}$. In summary, this analysis justifies the employment of our suggested EMA target prototypes.

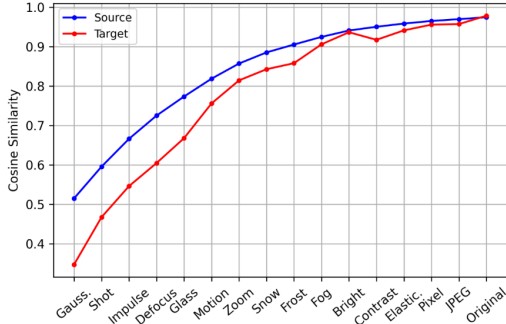

Figure 6: Cosine similarity analysis of $P^t$ with $P^s$ and $P^{t^*}$ for each target domain as the adaptation proceeds.

## I PREDICTION BIAS ANALYSIS OF EACH TARGET DOMAIN

In Fig. 5 (d), we compared the number of predicted samples per class between EATA and EATA+**Ours**, demonstrating that our proposed terms contribute to a more unbiased prediction of the model, encouraging the model to predict more diverse classes. Since Fig. 5 (d) shows the results summed over the all 16 domains, in Fig. 7, we break down the results by each domain and show the individual result of each domain. It is observed that the domains which the model shows high accuracy (brightness, original), also achieves a more balanced number of predicted samples per class across the classes. Conversely, in domains where the accuracy is low, we observe a significant bias in predictions, indicating that the model tends to favor certain classes excessively over others, making more frequent predictions on those classes. Overall, the bias is mitigated across all domains when our proposed terms are incorporated. EATA+**Ours** decreases predictions on the classes that EATA predicts frequently, instead, it increases predictions on the classes with a low number of predictions by EATA. Indeed, these findings confirm that our suggested terms effectively encourage the model to generate predictions that exhibit increased diversity among different classes. This mitigates the bias of the model towards favoring certain classes and, consequently, contributes to addressing the confirmation bias problem.

## J LIMITATION AND FUTURE WORK

Even though our proposed EMA target prototypical loss and source distribution alignment loss indeed contribute to significant performance improvement, there are some limitations to our work that can be further developed. The trade-off terms, $\lambda_{ema}$ and $\lambda_{src}$ for out proposed loss terms need to be fine-tuned depending on the specific method to which our proposed approach is applied. However, we have observed that it requires minimal effort to identify suitable values for these parameters, typically falling within the range of 1 to 2 for $\mathcal{L}_{ema}$ and 20 to 50 for $\mathcal{L}_{src}$. Since both $\mathcal{L}_{ema}$ and $\mathcal{L}_{src}$ rely on pseudo-labels for their computation, they can potentially result in the incorrect computation because pseudo-labels are not always accurate. To address this issue, we take measures to use only reliable samples for the computation of the loss terms. However, there is room for improvement in how we leverage pseudo-labels, such as refining them to be more precise or exploring alternative information sources for computing the loss terms.

Filtering out unreliable samples with high-entropy, is indeed an effective and efficient method to boost performance and enable efficient adaptation since it reduces the number of samples for adaptation by excluding unreliable samples. However, looking at it from a different perspective, if we can find ways to effectively harness these unreliable samples during test-time adaptation, they have the potential to make a substantial contribution to performance gains, as they represent challenging data that can introduce new insights. Disregarding high-entropy samples may inadvertently result in the loss of valuable information. Future research could focus on strategies to leverage the potential of these high-entropy samples and extract meaningful knowledge from them. We look forward to future research endeavors that aim to tackle the aforementioned limitations and explore the suggested avenues for future work.

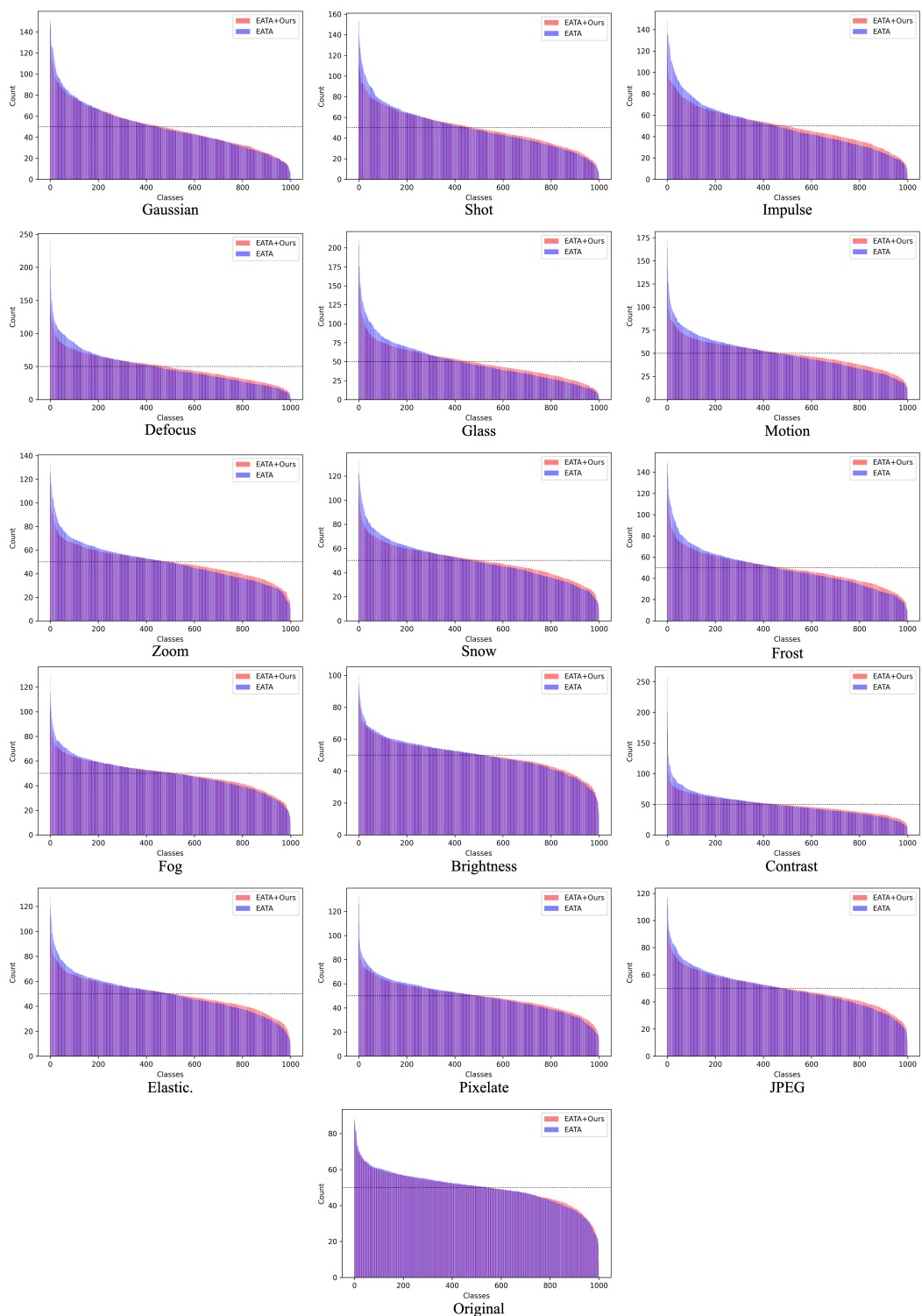

Figure 7: The comparison between EATA and EATA+**Ours** on the number of predicted samples per class.

