# OpenReview forum: "Continual Test-Time Adaptation by Leveraging Source Prototypes and Exponential Moving Average Target Prototypes"
_ICLR.cc/2024/Conference — Submitted to ICLR 2024_

### Official Review · Reviewer_hKy6 · 2023-10-25

**Soundness:** 2 fair
**Presentation:** 4 excellent
**Contribution:** 2 fair
**Rating:** 3
**Confidence:** 4

**Summary:**

The objective of this paper is to address the challenge of adapting a pre-trained Convolutional Neural Network (CNN) to distribution shifts during test time. These shifts stem from corrupted test images, which may include issues such as noise and blur.

To tackle this problem, the authors propose a method that penalizes abrupt changes in class prototypes by employing exponential moving average (EMA). By leveraging this technique, the authors aim to enhance the adaptability of the model in the face of distribution shifts caused by various forms of image corruption.

The authors conduct thorough evaluations of their approach on widely-acknowledged continual test time adaptation benchmarks, specifically Imagenet-C and CIFAR-100. Their results demonstrate that their method occasionally outperforms existing state-of-the-art test time adapters, namely EATA, CoTTA, and RMT.

Overall, the paper assembles a simple technique for addressing the challenges posed by distribution shifts, offering some insights into improving the adaptability of pre-trained CNNs under diverse test conditions.

---------------------------- Post Rebuttal --------------------------------------------------------------------------

I read through all the other reviews, as well as the rebuttal text. The rebuttal text re-approves the lack of technical contribution. I also notice that access to source data for adaptation is a major limitation (of the studied setting, not necessarily for this particular paper).

I keep my original score.

**Strengths:**

S1:
The paper addresses a crucial yet underexplored scenario: continuous adaptation to test time shifts.

S2:
The utilized approach is simple: it enforces gradual shifts in class prototypes instead of abrupt changes, achieved through the application of Exponential Moving Average (EMA).

S3:
The paper is well-written, and meticulously executed.

**Weaknesses:**

W1:
A significant concern regarding this paper is its lack of technical innovation. Despite being an application paper, the method merely applies EMA for continual test time adaptation, employing standard techniques for selecting reliable test exemplars, computing class prototypes, and calculating EMA penalties. These methodologies are well-established within the existing literature.

W2:
The empirical evidence presented in the paper lacks persuasiveness. A substantial performance boost could have justified the paper's simplicity and application-oriented nature. However, the minor improvement over EATA and RMT, as indicated primarily in Table 1-2 (i.e., less than +0.5% accuracy), does not substantiate the approach's effectiveness convincingly.

W3:
One notable omission in the paper is the absence of a comparison or reference against a significant baseline, namely NOTE [1], which is a robust continual test-time adaptation method designed to handle temporal correlations. This baseline is highly relevant to the authors' objectives and should have been included for a comprehensive evaluation. Furthermore, the paper missed an opportunity for greater depth by limiting its evaluation to simple, artificial distribution shifts induced by corruptions. It could have been more compelling if the authors had explored and evaluated the method against natural shifts or temporal correlations, thus enhancing the paper's overall impact and relevance.

**Questions:**

N/A

---

> ### Author Response · Authors · 2023-11-16
>
> W1 : Please refer to the official comment to all authors about the novelty of this work.
>
> W2 : We respectfully disagree with the assertion that our proposed method yields marginal performance gains. In the case of ImageNet-C, our proposed term enhances the mean accuracy across all target domains by more than 1% for all three applied methods (EATA, CoTTA, and RMT).  More than 1% performance gain is considered significant, especially given the challenging nature of the continual test-time adaptation setting and the dataset containing 1000 classes. Additionally, for CIFAR100-C, our proposed term leads to meaningful performance improvements for both EATA and CoTTA. The only scenario where marginal performance gains are observed is with RMT on CIFAR100-C, attributed to its intricate incorporation of several loss terms.
> We are uncertain about the reason for stating that there is an accuracy gain of less than +0.5%.
>
> Moreover, we conducted through ablation study and analysis to validate the efficacy of our proposed method. We respectfully request the reviewer to check our ablation study and analysis as well.
>
> W3 : NOTE addresses the temporal correlation of TTA, primarily focusing on the imbalanced class distribution of incoming target data. While this differs slightly from our problem, which specifically addresses the domain shift of target data, we plan to incorporate NOTE as one of the baseline methods in the revised version. Additionally, we will explore other datasets that closely resemble the natural shifts of continual test-time adaptation setting.

---

### Official Review · Reviewer_QiZ6 · 2023-10-28

**Soundness:** 3 good
**Presentation:** 2 fair
**Contribution:** 2 fair
**Rating:** 5
**Confidence:** 4

**Summary:**

This work tackles the continual test-time adaptation problem. It proposes two enhancements that are orthogonal to several test-time adaptation methods. The first component as regularizing the feature extractor with a cross-entropy loss that leverages a moving average of  prototypical features from the target domain. Such features are initialized with the weights of the linear classifier and then updated using exponential moving average. The second component is to align the features of the target domain with a precomputed set of prototypical features from the source domain. Experiments are carried out on two datasets (CIFAR-100-C and ImageNet-C) where the proposed method showed performance gains when combined with 3 test-time adaptation methods.

**Strengths:**

The main strengths of this work are:

- The problem this paper tackles is both important and practical.

- The approach proposed in this work is simple and easy to implement. Further, the experiments show the applicability of the proposed enhancements when combined with different test-time adaptation methods.

- The online estimates of the prototypical features and the MSE loss makes the proposed approach efficient as demonstrated in Figure 2.

**Weaknesses:**

Despite the stated strengths of this work, there are several weaknesses that need to be addressed before accepting this work.

1- Methodology. While the proposed method is simple to both understand and implement, there are several caveats that need to be discussed:

(1a) How are the hyper parameters tuned? Is the source data used to initialize $P^s$ employing training data or validation data?

(1b) This paper needs to properly state its contributions over TTAC (the other clustering approach in the literature).

(1c) It is unclear whether the predictions in Algorithm 1 line 5 are adjusted before the output phase (line Ensure) as the predictions $z$ are returned as $\hat y$. Further, the algorithm returns the set of predictions for all data-points and all domains rather than conducting the evaluation in an online manner (return the predictions batch by batch).

2- Experiments. The experimental analysis in the work show marginal performance gains of the proposed approach. Further, there are missing key experimental details and comparisons:

(2a) How is the hyperparemeter search done for the proposed approach? Is a similar effort put into other baselines (e.g. EATA + TTAC)?

(2b) It is unclear why the proposed components degrade the performance of EATA under small batch-sizes?

(2c) While EATA does not regularize for its features to be clustered (unlike the proposed approach), they are still very competitive (better in discriminating different clusters) when the proposed components are absent.

(2d) Generally, I think the analysis and comparison of the proposed method should be against EATA+TTAC rather than EATA (e.g. in Figure s 4 and 5).

(2e) Experiments with different and more powerful architectures that do not use batch normalization (e.g. ViT) layers are missing.

3- Writing. The writing of this work should be vastly improved to enhance its readability. Here are a list of suggestion to be considered in the final version:

(3a) The problem definition is not clear. Both $k$ and $t$ are used to refer to time.

(3b) The introduction should state clearly the contributions this work provides.

(3c) Algorithm 1 is unclear how the prediction is conducted and what predictions are returned (line 5 computes z while last line returns $
\hat y$.

(3d) Captions of Tables 3 and 4 should be improved to elaborate what metric is reported and what to pay attention to.

(3e) The related work section is missing from the main paper and put in the appendix. It is essential to have this section to clearly position this work in the literature.

Overall, while I like the simplicity of the proposed approach, the performance gains are generally very marginal (with best choice of hyper parameters) questioning the usefulness of the proposed method.

**Questions:**

Please refer to the weaknesses section.

---

> ### Author Response · Authors · 2023-11-16
>
> (1a) Hyper-parameters are tuned via grid search. Source data used to initialize $P^s$ are sub-samples of training data. It is stated in the section 3.2 of our original submission.
>
> (1b) TTAC assumes that each class cluster follows a Gaussian distribution. TTAC pre-calculate the mean and covariance ($\mu_s \in \mathbb{R}^d$ and $\Sigma_s \in \mathbb{R}^{d \times d}$, where $d$ is the dimension of the feature) for each class of source domain data, computing Gaussian distributions for each source class prior to test-time adaptation. At test-time, it uses the target data to compute the Gaussian distribution for each class cluster of target domain as well. It uses the distributions of source domain class clusters as anchors for the distributions of target domain class clusters to match against. The anchoring of cluster is achieved by directly minimizing the KL-Divergence between the source and the target distributions.
>
> Computing Gaussian distribution for each class cluster and minimizing KL-Divergence between the two distributions require computation overhead. We show that it indeed requires significant time to adapt a single batch in Figure 2 while our proposed method requires a small amount of time for adaptation. Pleas refer to the difference between the green and the blue plot of Figure 2.
> We also show that in the main performance comparison tables (Table 1 and Table 2), ours outperforms TTAC in both benchmarks, showing superiority over TTAC in both accuracy and adaptation time.
>
> Unlike TTAC, our proposed method does not require the computation of intricate Gaussian distributions or the KL-Divergence between source and target distributions. Instead, it directly minimizes the MSE distance between the target feature and the source prototypes ($P^s$), akin to $\mu_s$ in TTAC. This makes our method simpler, more efficient, and achieves superior performance over TTAC when combined with $\mathcal{L}_{ema}$.
>
> (1c) Our algorithm returns the predictions batch by batch in an online manner. Line "Ensure" indicates that our algorithm ensures the prediction outputs for all the inputs from all the target domains. It does "not" mean that the predictions are returned all together at "once" when the test-time adaptation is finished.
>
>
> (2a) As addressed in response to question (1a), we employed grid search to determine the hyperparameters. Reference to Figure 3 in the analysis section provides additional insights. As outlined in the implementation details in Appendix B, we adhered to the hyper-parameters specified in the official code or the original paper of baseline methods. Nevertheless, we dedicated ample effort to fine-tune the hyper-parameters to attain satisfactory performance within the given CTA setting.
>
> (2b) When employing small batch sizes, the model exhibits low performance in batch accuracy. For a majority of batches (iterations), the accuracy is either 0% or less than 50%. This low accuracy hampers the construction of EMA target prototypes with accurate pseudo-labels, as well as the source alignment loss. Furthermore, given the reduced number of reliable samples after entropy-based sample selection due to low accruacy, the target prototypes are infrequently updated and remain unchanged. These challenges result in diminished performance of our proposed loss terms when using small batch sizes.
>
> (2c) EATA uses source samples to compute fisher importance of model weights before performing test-time adaptation. With the computed fisher importance, it employs anti-forgetting loss which prevents the parameters of the model from changing too much from their source trained weights during the test-time adaptation. It also selects reliable and non-redundant samples to facilitate efficient entropy minimization loss.
>
> (2d) The analysis in Figure 4 and 5 are conducted between EATA and EATA+Ours to show the efficacy of our proposed terms. We wanted to analyze how our proposed terms affect the model by ablating the proposed terms. If we compare between EATA+TTAC and EATA+Ours, we can not properly analyze how our proposed method has effect on the model.
>
> (2e) We will try to include experiments using different network architecture in the future version.
>
> (3a) $k$ refers to the domain index as mentioned in the paper while $t$ refers to the training iteration.
>
> (3b) Our contributions are well stated in the last paragraph of the introduction.
>
> (3c) Please refer to our answer on (1c).
>
> (3d) The metric presented in Table 3 is the average classification accuracy across the 16 domains, computed over three separate runs, and reported with standard deviation. The metric of Table 4 is the same as Table 1 and 2. What to pay attention to is well explained in each section, but we will try to briefly explain them in the caption for later version.
>
> (3e) We will include the related work in the main paper in future version.

---

> > ### Author Response · Authors · 2023-11-16
> >
> > We respectfully disagree with the assertion that our proposed method yields marginal performance gains. In the case of ImageNet-C, our proposed term enhances the mean accuracy across all target domains by more than 1% for all three applied methods (EATA, CoTTA, and RMT).  More than 1% performance gain is considered significant, especially given the challenging nature of the continual test-time adaptation setting and the dataset containing 1000 classes. Additionally, for CIFAR100-C, our proposed term leads to meaningful performance improvements for both EATA and CoTTA. The only scenario where marginal performance gains are observed is with RMT on CIFAR100-C, attributed to its intricate incorporation of several loss terms.

---

### Official Review · Reviewer_iXmR · 2023-10-31

**Soundness:** 2 fair
**Presentation:** 2 fair
**Contribution:** 2 fair
**Rating:** 5
**Confidence:** 5

**Summary:**

The paper introduces a method that utilizes prototypes from both source and target domains to enhance continual test-time adaptation. This approach seamlessly integrates with existing CTA methods, using source prototypes to reduce distribution discrepancies and target prototypes to cluster target features, which are updated via an EMA process during test-time. The research showcases improved model performance, reduced adaptation time overhead, and a mitigation of model bias, paving the way for future advancements in continual test-time adaptation techniques.

**Strengths:**

**Originality**: The paper innovatively combines prototypes from both source and target domains, offering a fresh perspective on continual test-time adaptation.

**Quality**: The rigorous methodology and in-depth experiments validate the efficacy of the proposed terms, demonstrating tangible improvements in model performance.

**Clarity**: The paper is well-structured and articulates its methods and findings with precision, making it accessible for readers familiar with the domain.

**Significance**: By addressing model bias and enhancing adaptation efficiency, this research holds potential to shape future work in the realm of continual test-time adaptation.

**Weaknesses:**

(1) Compared to CoTTA, where unsupervised test-time adaptation methods function without relying on source domain data, this paper's reliance on pre-computed source prototypes from the source domain data seems to be an uneven ground. It raises questions about the comparability and fairness of the presented method relative to others, like CoTTA, which function without such dependencies.

(2) The various loss components highlighted in the paper have previously been discussed in many other studies within the Domain Adaptation (DA) and Test-Time Adaptation (TTA) fields. This gives an impression that the proposed method might just be a combination of existing techniques, akin to piecing together different methods like A+B+C. Additionally, the usage of $L_{ema}$, which is commonly found in unsupervised DA, doesn't appear novel.

(3) Figure 1 in the paper comes across as overly intricate and disorganized. A reader would need to invest significant effort to discern and correlate the various methods depicted, which hampers the immediate understanding of the paper's methodology.

(4) The $L_{unsup}$ seems to be inadequately defined in the paper. Even though the authors touched upon it in the "Problem definition" section, a more explicit formula or representation would have been beneficial for clarity and a more straightforward comprehension.

**Questions:**

1. **Regarding Comparability:** One of the primary concerns raised revolves around the fairness of comparing the proposed method with traditional CTA methods. Given that the proposed method relies on pre-computed source prototypes from the source domain data, how do the authors justify the comparability of their method, especially when other methods like CoTTA operate without such dependencies?

2. **Concerning Novelty:** The various loss components presented in the paper seem to have been discussed in previous DA and TTA research. Could the authors elaborate on what sets their method apart, particularly concerning its novelty? How does the incorporation of these loss components enhance the uniqueness and effectiveness of the proposed method?

3. **On Clarity:** Figure 1, as mentioned, appears quite intricate. Is there a possibility to streamline or restructure the figure to make it more intuitive for readers?


If the authors can address these questions and take into consideration the suggestions provided, it would greatly enhance the paper's clarity and relevance, and I will consider increasing my score.

---

> ### Author Response · Authors · 2023-11-16
>
> **Regarding Comparability**
>
> As we addressed in response to reviewer JC3N, it is crucial to highlight that baseline methods like TTAC, EATA, RMT all leverage source domain data according to their proposed methodologies. Nevertheless, we acknowledge the valid point raised by you, suggesting that comparing with methods that do not utilize any source domain data, such as CoTTA, may appear unfair.
>
> In the table below, we show that even in the absence of source domain data, specifically without employing $\mathcal{L}_{src}$
>
> and solely utilizing $\mathcal{L}_{ema}$, there is a performance improvement.
>
> This table shows the average accuracy over the 16 test domains of ImageNet-C benchmark.
>
> |Method|Mean Acc. over the 16 test domains.|
> |----|----|
> |$\mathcal{L}_{ema}$|44.00%|
> |$L_{ema} + \mathcal{L}_{src}$|45.96%|
> |EATA|49.81%|
> |EATA+$\mathcal{L}_{ema}$|50.62%|
> |EATA+**Ours**|51.32%|
> |CoTTA|37.42%|
> |CoTTA+$\mathcal{L}_{ema}$|46.39%|
> |CoTTA+**Ours**|46.91%|
> |RMT|41.68%|
> |RMT+$\mathcal{L}_{ema}$|42.29%|
> |RMT+**Ours**|42.78%|
>
> Notably, when $\mathcal{L}_{ema}$  is applied independently of the baseline method, it already outperforms CoTTA (44.00% vs. 37.42%).
>
> Also, CoTTA+$\mathcal{L}_{ema}$ shows remarkable performance compared to CoTTA, 
>
> supporting the validity of proposed $\mathcal{L}_{ema}$.
>
> $\mathcal{L}_{src}$ can be additionally employed to further boost the performance when a sub-sample of source domain data are available.
>
> It is essential to underscore that CoTTA necessitates additional computation and memory resources due to its teacher-student framework and augmentation strategy.
>
> This results in a longer adaptation time for a single batch, as illustrated in Figure 2 of our original submission.
>
> **Concerning Novelty**
>
> Please refer to the official comment to all authors about the novelty of this work.
>
> **Clarity of Figure 1**
>
> We will elaborate the Figure 1 for better understanding of the proposed methodology in the future version.
> We consider having a sperate figure for each proposed loss term and further describe each loss term in more detail.
>
>
> **Formulation of $\mathcal{L}_{unsup}$**
>
> As stated in the problem definition, $\mathcal{L}_{unsup}$ can take a form of entropy minimization loss, akin to the approach in TENT and EATA. Alternatively, it can adopt the form of a distillation loss from the teacher network to the student network, as employed in CoTTA.
>
> Entropy minimization loss term is defined as follows:
>
> $$
> \mathcal{L}_{ent}=-\sum^C_c (\sigma(g(x^t))_c \cdot \log(\sigma(g(x^t))_c))
> $$
>
> Where $\sigma$ is the softmax operation and  $g_{\theta}(x^t)_c$
>
> refers to the $c$-th element of the produced logit $z^t=g_{\theta}(x^t) \in \mathbb{R}^C$.
>
> It is basically a loss to minimize the entropy of the produced logit.
>
> Distillation loss term (or the consistency loss as named in CoTTA) is defined as follows:
>
> $$
> \mathcal{L}_{ent}=-\sum^C_c(\sigma(g_t(x^t))_c \cdot \log(\sigma(g_s(x^t))_c))
> $$
>
> Where $g_{t}$ and $g_{s}$ refer to the teacher and the student networks.

---

### Official Review · Reviewer_JC3N · 2023-11-01

**Soundness:** 3 good
**Presentation:** 3 good
**Contribution:** 2 fair
**Rating:** 6
**Confidence:** 3

**Summary:**

This paper utilizes the idea of prototypes for the problem of continual test-time adaptation.
The proposed approach precomputes class prototypes for the source domain data and uses these to perform prototype matching with the target prototypes.
The target domain data is used to compute the target prototypes with the features of only the reliable low entropy samples.
Using the target samples provided at test time, the target prototypes are updated using the exponential moving average (EMA).
Experimental results on ImageNetC and CIFAR100C suggest the effectiveness of the proposed approach.

**Strengths:**

* Utilizing the prototypical learning approach for test-time adaptation
* Extensive ablation study to study the effect of EMA weight and different components on the overall objective

**Weaknesses:**

* Computating the source prototypes requires the source domain data. So, the proposed approach will not work for any off-the-shelf pre-trained model without the source domain data
* Utilizing class prototypes is limited to classification tasks and not generalizable to other tasks
* Experiments are limited to ImageNetC and CIFAR100C, and CIFAR10C related experiments are missing

**Questions:**

1. Can the authors report the experimental results on CIFAR10C, since prior works report their performance for this benchmark?
2. Utilizing class prototypes is limited to classification tasks and not generalizable to other tasks. Can this approach be generalizable to other tasks, such as segmentation?

---

> ### Author Response · Authors · 2023-11-16
>
> **The usage of source domain data**
>
> First, we want to mention that our baseline methods also require source domain data. TTAC requires computing source cluster parameters ($\mu_s \in \mathbb{R}^d$ and $\Sigma_s \in \mathbb{R}^{d \times d}$, where $d$ is the dimension of the feature), the mean and covariance, with the source data prior to test-time adaptation. EATA requires source domain data to compute the importance weights for the anti-forgetting regularization before deploying the model to the target domains. RMT also employs source replay loss which accesses the source domain data during test-time adaption.
> CoTTA, TSD, and other methods do not necessitate source domain data, yet they exhibit comparatively lower performance.
>
> In the table below, we show that even in the absence of source domain data, specifically without employing $\mathcal{L}_{src}$
>
> and solely utilizing $\mathcal{L}_{ema}$, there is a performance improvement.
>
> $\mathcal{L}_{src}$ can be further incorporated into training the model when there are some source samples available.
>
> This table shows the average accuracy over the 16 test domains of ImageNet-C benchmark.
>
> |Method|Mean Acc. over the 16 test domains.|
> |----|----|
> |$\mathcal{L}_{ema}$|44.00%|
> |$L_{ema} + \mathcal{L}_{src}$|45.96%|
> |EATA|49.81%|
> |EATA+$\mathcal{L}_{ema}$|50.62%|
> |EATA+**Ours**|51.32%|
> |CoTTA|37.42%|
> |CoTTA+$\mathcal{L}_{ema}$|46.39%|
> |CoTTA+**Ours**|46.91%|
> |RMT|41.68%|
> |RMT+$\mathcal{L}_{ema}$|42.29%|
> |RMT+**Ours**|42.78%|
>
> **CIFAR10-C experiments**
>
> We did not conduct experiments on CIFAR10-C, since CIFAR100-C and ImageNet-C are more difficult tasks with a larger number of classes which are adequate benchmarks to show the validity of our proposed method.
> However, we plan to include results on CIFAR10-C in future version.
>
> **Generalization to other tasks**
>
> While certain specific implementations need further development, we believe that our proposed method holds the potential for generalization to other tasks, such as semantic segmentation.
> Given that segmentation involves pixel-wise predictions, the proposed loss terms $\mathcal{L}_{src}$
>
> and $\mathcal{L}_{ema}$ can be applied in a pixel-wise manner.
>
> We can construct the source prototypes and the EMA prototypes with the features given prior to the prediction head of the segmentation network.
> We can collect reliable pixel-wise features exploiting the prediction results from the segmentation head, then we can aggregate them to generate the proposed prototypes.
> Then, with the generated prototypes, we can compute the loss terms $\mathcal{L}_{src}$
>
> and $\mathcal{L}_{ema}$ on predictions given by the segmentation network.

---

> > ### Comment · Reviewer_JC3N · 2023-11-22
> > **Authors' response Acknowledgement**
> >
> > Thanks to the authors for responding to the queries and providing additional experiments.
> >
> > The numbers in the absence of source domain data seem to suggest a marginal improvement for the proposed approach (+Ours).
> >
> > Regarding generalizing to other problems, even to construct the source and EMA prototypes with the features prior to the prediction head, one would need a notion of class. So, it still seems non-trivial to me as to how to address TTA with the proposed approach for problems where there is no notion of class.
> >
> > Currently, I will keep my score unchanged.

---

> > > ### Author Response · Authors · 2023-11-22
> > >
> > > Thanks for your response.
> > >
> > > We think the notion of class is inevitable. Without the information of classes, we can not even define the output space of the classifier (the last linear layer). TTA assumes that the number of classes and their information are given and only deals with the domain shifts problem.

---

### Official Review · Reviewer_eihD · 2023-11-01

**Soundness:** 3 good
**Presentation:** 2 fair
**Contribution:** 2 fair
**Rating:** 5
**Confidence:** 4

**Summary:**

This paper studies the problem of continual test-time model adaptation, where training data is not accessible and only continually changing target domains are available. To keep adapting the model to the continually changing target domains in an online manner, they propose to maintain an exponential moving average target prototype for each class with reliable target samples. In addition, semantic alignment is achieved by matching the target feature to its corresponding pre-computed source prototype. Experiments on standard benchmarks demonstrate the effectiveness of the proposed approach.

**Strengths:**

- The problem of addressing continuously shifted target domains is a significant yet under-explored topic, especially in the context of test-time adaptation.

- The proposed prototype alignment with EMA target prototypes is simple and easy to implement.

- The experiments conducted in the manuscript provide a comprehensive comparison with the most closely related works, spanning across a broad array of Test-Time Adaptation (TTA) benchmarks and baseline methods.

**Weaknesses:**

- My major concern is about the novelty. Prototypical alignment has been extensively explored in previous domain adaptation methods. The major difference between the proposed approach and those prior efforts seems marginal. I suggest that the authors provide detailed comparisons and showcase why their proposed approach is preferable when applied to test-time adaptation scenarios.

- The writing quality of the article is average, with unclear logical progression and lack of fluency in some parts. There are also numerous instances of imprecise word usage. I recommend that the authors have a native English speaker conduct a thorough proofreading to enhance the clarity, coherence, and overall readability of the manuscript. This will ensure that the paper meets the high standards expected of publications in this field and effectively communicates its contributions to the intended audience.

**Questions:**

Please refer to the weaknesses.

---

> ### Author Response · Authors · 2023-11-16
>
> **The novelty of the work**
>
> Please refer to the official comment to all authors about the novelty of this work.
>
> **Writing quality**
>
> Regarding the writing quality, it would be greatly beneficial and constructive if you could pinpoint specific sections and sentences in our paper that do not meet the standards you have mentioned. Your feedback will enable us to enhance the overall writing quality in future version, and we appreciate any guidance you provide in this regard. We also respectfully ask you to consider the reviews of reviewer iXmR and hKy6 that the paper is well-structured and written.

---

### Author Response · Authors · 2023-11-16
**Regarding the novelty of the work-(1)**

First of all, we express our great gratitude for the time and effort reviewers dedicated to reviewing our paper.

Several authors stated that they are concerned about the novelty of our work. However, we are concerned that none of the reviewers mentioned any specific names of previous works that share similar methodologies. We would highly appreciate it if you can share the specific references of the previous methods that show similar approaches to our work so that we can explain how our work is different in more detail. Pointing out novelty without specific references is unconvincing and not desirable.

As mentioned in our introduction, the biggest hurdle in continual test-time adaptation is that the distribution of the target domain changes as time goes on.
Due to its intricate setting of changing data distributions, the model may abruptly encounter different target domains and forget knowledge learned from the source domain while adapting to the target domain in an online manner.

To mitigate the issue, we propose the use of exponential moving average (EMA) target prototypes ($P^t$). These target prototypes are initialized as the weights of the classification head, $\omega \in \mathbb{R}^{C \times d}$. Since the weights are trained using the source domain with ground truth labels prior to test-time adaptation, each weight vector, $\omega_c \in \mathbb{R}^d$, represents the prototype of each class cluster.

Then, as the test-time adaptation proceeds, they are continuously updated with the features of the target domain inputs given at test-time via Eq.(2).
The goal of the EMA target prototypes is to constantly capture the changing target distributions in an EMA manner to maintain a prototype for each class that generalizes well across different target domains.

By exploiting the evolving EMA target protototypes, we aim to discriminate each feature of target input into distinct classes in the target feature space. Therefore, we propose $\mathcal{L}_{ema}$ to anchor each target feature to its corresponding EMA target prototype, ensuring separation from other unrelated EMA target prototypes.

In short, Eq.(2) is to maintain class-wise prototypes that reflect changing target distributions in an EMA manner, while Eq.(1) is to exploit these prototypes to organize the given target features by classes.

When initializing and updating the EMA target prototypes, we normalize the head weights $\omega$ and the extracted target feature $f_{\phi}(\tilde{x}^t)$ to eliminate the magnitudes. We observe that the normalization is important as its ablation leads to lower performance in the ablation study.
We see the normalization as an novel property that makes our $\mathcal{L}_{ema}$ differentiated.

Also, $\mathcal{L}_{ema}$ enables to quickly capture the changing distribution of streaming target data in an online fashion without relying on memory bank or additional learnable parameters. Not requiring any learnable parameter or memory makes this approach very simple to implement and efficient to use under restricted environment with limited resources such as power, computation and memory.

Moreover, our proposed source alignment loss $\mathcal{L}_{src}$ ensures to reduce the domain gap by anchoring each feature of target input to its corresponding source prototype ($P^s$).

The proposed approach proves to be favorable and effective in the continual test-time adaptation setting since it facilitates the model in ongoing learning of the changing target domain data by leveraging the EMA target prototypes, all the while narrowing the domain gap and preventing the loss of source domain knowledge by incorporating the source prototypes throughout the test-time adaptation without significant computation overhead.

As outlined in the paper, we include TSD and TTAC as our baseline methods to compare, despite their original designs not specifically targeting continual test-time adaptation, due to the shared similarities with our approach.

In contrast to TSD, our proposed method operates without the use of memory.
Additionally, unlike TTAC, our approach eliminates the need for computing the Gaussian distribution for each class cluster of both the source and the target domains and minimizing the KL-Divergence between the Gaussian distributions of the source and target domains.

Instead, our proposed method directly minimizes the mean squared error (MSE) distance between the target feature and the corresponding source prototype ($P^s$) and incorporate $\mathcal{L}_{ema}$ for further clustering of classes at target feature space.

This makes our method simpler, more efficient, and achieves superior performance in adaptation time over both TSD and TTAC, as illustrated in Figure 2.

Our proposed method may have some similarities with existing methods, but it is different in details and our experimental results showcase that our method surpasses both TSD and TTAC in terms of both accuracy and adaptation time.

---

> ### Author Response · Authors · 2023-11-16
> **Regarding the novelty of the work-(2)**
>
> We posit that the novelty of a work is not solely derived from the technical innovations it introduces but also from the experimental results and analysis presented within the work.
> Our proposed method may have some similarities with approaches explored in previous methods, but to the best of our knowledge, it has not been heavily studied under the continual test-time adaptation scenario.
> Therefore, we believe it is worth studying how it affects the model and solves the problem with extensive experiments and analysis under the CTA scenario.
>
> In the analysis section, we have provided various analysis experiments on how our proposed method affects the model and its output.
> We observed that our proposed term mitigates bias in the model, addressing the tendency to favor certain classes more frequently than others.
> It also effectively narrows down the source-target domain gap and reduces intra-class distance.
> We earnestly ask reviewers to consider this for the final decision.

---

### Meta-Review · Area_Chair_cB5q · 2023-12-02

**Metareview:**

This paper studies the problem of continual test-time model adaptation. Specifically, the authors propose to maintain an exponential moving average target prototype for each class with reliable target samples. In addition, semantic alignment is achieved by matching the target feature to its corresponding pre-computed source prototype. Experiments on standard benchmarks demonstrate the effectiveness of the proposed approach.

**Strengths**

- This paper studies a significant yet under-explored topic.

- The proposed prototype alignment with EMA target prototypes is simple and easy to implement.

- The experiments conducted in the manuscript provide a comprehensive comparison with the most closely related works

- The combination of  prototypes from both source and target domains provides a fresh perspective on continual test-time adaptation.

- The paper is well-structured and easy to follow.

**Weaknesses**

- The main concern is the limited novelty agreed by most of the reviewers. Prototypical alignment has been extensively explored in previous domain adaptation methods. And, the proposed method is more like a combination of existing techniques.

- In some cases, the improvements are somewhat not significant.

**Justification For Why Not Higher Score:**

The novelty of the proposed is somewhat limited or should be further clarified compared to existing methods.

**Justification For Why Not Lower Score:**

N/A

---

### Decision · Program_Chairs · 2024-01-16

Reject